# Proteins with proximal-distal asymmetries in axoneme localisation control flagellum beat frequency

Cecile Fort [1,4], Benjamin J. Walker [2,3], Lore Baert[1,5] & Richard J. Wheeler [1,6] ✉

The 9 + 2 microtubule-based axoneme within motile flagella is well known for its symmetry. However, examples of asymmetric structures and proteins asymmetrically positioned within the 9 + 2 axoneme architecture have been identified. These occur in multiple different organisms, particularly involving the inner or outer dynein arms. Here, we comprehensively analyse conserved proximal-distal asymmetries in the uniflagellate trypanosomatid eukaryotic parasites. Building on the genome-wide localisation screen in *Trypanosoma brucei* we identify conserved proteins with an analogous asymmetric locali-sation in the related parasite *Leishmania mexicana*. Using deletion mutants, we find which are necessary for normal cell swimming, flagellum beat parameters and axoneme ultrastructure. Using combinatorial endogenous fluorescent tagging and deletion, we map co-dependencies for assembly into their normal asymmetric localisation. This revealed 15 proteins, 9 known and 6 novel, with a conserved proximal or distal axoneme-specific localisation. Most are outer dynein arm associated and show that there are multiple classes of proximal-distal asymmetry – one which is dependent on the docking complex. Many of these proteins are necessary for retaining the normal frequency of the tip-to-base symmetric flagellar waveform. Our comprehensive mapping reveals unexpected contributions of proximal-specific axoneme components to the frequency of waveforms initiated distally.

Flagella and motile cilia are microtubule-based organelles used for motility and found across diverse eukaryotic lineages. They share a highly symmetric core architecture, known as the axoneme, which is based on nine doublet microtubules around a central pair of singlet microtubules. However, asymmetries in the distribution of axoneme-decorating proteins are emerging as being important for the function of cilia and flagella[1].

Flagella employ different waveforms specific to their function: typically, a planar symmetric near-sinusoid flagellar-type beat (e.g. human sperm) or a planar asymmetric ciliary-type beat (e.g. ciliated epithelia). Some flagella can switch waveform type, with *Chlamydomonas* adopting an asymmetric beat for normal swimming and a symmetric beat during a bright-light phototactic response[2–4], in this case maintaining a base-to-tip direction of waveform propagation. Other examples can also switch waveform direction between base-to-tip and tip-to-base (e.g. *Leishmania*)[5,6]. *Leishmania* are one of the trypanosomatid parasites, a family which also includes *Trypanosoma brucei*. Due to tractable reverse genetics, these uniflagellate human pathogens form a powerful model system for analysing flagellar biology[7,8]. Trypanosomatid parasite

[1]Medawar Building for Pathogen Research, Nuffield Department of Medicine, University of Oxford, Oxford, UK. [2]Department of Mathematical Sciences, University of Bath, Bath, UK. [3]Department of Mathematics, University College London, London, UK. [4]Present address: Diamond Light Source, Didcot, UK. [5]Present address: Swiss Tropical and Public Health Institute, University of Basel, Basel, Switzerland. [6]Present address: Institute of Immunology and Infection, School of Biological Sciences, University of Edinburgh, Edinburgh, UK. ✉e-mail: r.wheeler@ed.ac.uk

flagellum-driven motility is necessary for normal life cycle progression[9,10].

The flagellar beat is generated by the action of dynein motor protein complexes attached to the axoneme. The outer dynein arms (ODAs) are attached to the nine doublets with a regular spacing of 24 nm and are canonically viewed as being the primary driver of the flagellar beat[11]. Contrastingly, there are multiple different inner dynein arm (IDA) complexes attached in a larger 96 nm[12] repeating unit, viewed as necessary for controlling the flagellar beat waveform rather than generating it[11]. However, recently, we showed that the preferred direction of waveform propagation in trypanosomatids is associated with a linear proximal-distal asymmetry in the ODAs[1]. We previously identified two paralogues of the ODA-docking complex (DC) heterodimer[13], one heterodimer specific to the proximal (pDC) and one to the distal (dDC) axoneme[1]. Deletion of dDCs lead to loss of the distal ODAs, which caused decreased normal tip-to-base symmetric beating and increased the rarer base-to-tip asymmetric beat. While this precise DC-dependent asymmetry appears specific to the trypanosomatids, analogous DC or ODA asymmetries are found in diverse eukaryotes, including the unicellular parasite *Giardia*, the green alga *Chlamydomonas* and Humans[1,14–19].

This is not the only potentially conserved proximal-distal asymmetry. ARL13B is a well-characterised conserved marker of cilia and flagella necessary for normal ciliary/flagellar length, which localises to the full length of human cilia[20–27]. However, in some cases, including mouse oviduct and tracheal tissue, ARL13B, detected as a eGFP fusion protein, is enriched in the proximal axoneme[28]. A similar proximal localisation is seen in *T. brucei* by endogenous fluorescent protein tagging and anti-ARL13B immunofluorescence[27]. Phosphodiesterases (PDEs) have also been identified as enriched in the distal flagellum: PDEA in *L. mexicana*[9] and PDEB in *T. brucei*[29,30]. This suggests complexity in proximal-distal axoneme composition beyond the DCs.

Cyclic AMP (cAMP) and calcium ion ($Ca^{2+}$) signalling are often implicated in the control of beat type to control cell motility[31–37]. For example, $Ca^{2+}$ signalling is involved in the phototactic response of *Chlamydomonas*[38–40]. Trypanosomatid parasites are no exception: $Ca^{2+}$ and cAMP alter beat type of demembranated reactivated *Leishmania* axonemes[41] and there are likely links to asymmetrically positioned proteins. FLAM6 in *T. brucei*[42], has a proximal DC-like localisation and has a predicted cAMP binding domain, flagellar cAMP signalling by PDEB is necessary for productive infection of the *T. brucei* insect vector[43], and we previously identified LC4-like which has a predicted $Ca^{2+}$ binding domain as a distal axoneme-specific ODA-associated protein necessary for normal beat frequency in *T. brucei* and *Leishmania*[1]. Overall, this suggests an interplay of signalling and asymmetric protein distribution.

It is becoming clear that complex proximal-distal asymmetries exist in the axoneme, and there are hints that proteins with a proximal-distal asymmetry in axoneme localisation may function in flagellum beat control. We therefore sought to map, genome-wide, all conserved proximal-distal asymmetries in a model flagellum. Using genome-wide subcellular protein localisations in *T. brucei*, we identified all proximal- and distal-specific proteins and chose the subset with a *L. mexicana* ortholog and an analogous sub-axonemal localisation. We show that there are at least two distinct proximal and distal asymmetries. We discover the cohort which are dependent on the DC or/and ODA heavy chains for their localisation. This included identifying a new paralogous pair of DC proteins (one is proximal and one distal) necessary for normal ODA assembly. Finally, we demonstrated that a subset of proximal and distal-specific proteins are necessary for normal control of flagellum beating, including, surprisingly, that proximal-specific proteins can influence frequency of waveforms starting at the distal tip of the flagellum.

## Results

### Fifteen proteins have proximal-distal asymmetry conserved between *T. brucei* and *L. mexicana*

To comprehensively identify proximal- and distal-specific proteins that may be responsible for flagellum beat control, we used the TrypTag (genome-wide subcellular protein localisation in *Trypanosoma brucei*) dataset[44]. Through a manual survey of all proteins annotated with a flagellum or axoneme localisation, we selected proteins with a proximal- or distal-specific localisation by N or C terminal tagging at the endogenous locus, excluding proteins localising to a known flagellar sub-compartment (ie. transition zone, flagellum tip) or with signal enrichment (stronger, but not exclusive) rather than signal specific to the proximal or distal axoneme. This left 51 proximal- or distal-specific flagellar proteins (Fig. 1A, B, Figure. S1–3, Supplementary Data 1), and most localised by both N and C terminal tagging had very high similarity. A large majority of axoneme-localising proteins, including all conserved proteins in axonemal structures like the inner and outer dynein arms, radial spokes, nexin links/dynein regulatory complex and central pair complex, with the exception of the DC[1], are absent from this set as they do not have a proximal or distal-specific localisation.

We reasoned that proximal or distal-specific proteins with an *L. mexicana* ortholog that also has a proximal or distal-specific localisation are most likely to be functionally important, therefore we selected them for detailed analysis (Supplementary Data 1). To determine if the proximal- or distal-specific localisation was conserved between *T. brucei* and *L. mexicana*, we tagged the *L. mexicana* orthologs with mNeonGreen (mNG) at either the C or N terminus at their endogenous loci, selecting a terminus based on successful endogenous tagging in *T. brucei* (Fig. 1C, D). Overall, ~50% (24/51) of *T. brucei* proximal or distal-specific proteins have an *L. mexicana* ortholog, of which ~60% (15/24) have a comparable asymmetric localisation (Fig. 1A–D) while the remainder do not (Fig. S1), leaving 15 proteins for detailed analysis.

Visual inspection of the sub-flagella localisation in these two species suggested that there were several classes of asymmetric distribution. To characterise this, we measured the fluorescence signal distribution along the axoneme for all *T. brucei* asymmetrically localised proteins (Fig. S1A, B, S1, S2, S3). Hierarchical clustering, in comparison to two radial spoke proteins which localise to the full flagellum length[45] as controls, suggests numerous clusters. Although this approach cannot define the number of clusters, we have marked key groupings to guide interpretation (Fig. 2A). Where successfully generated, both N and C terminal tagging of the same protein typically fell within the same group (15/25, Supplementary Data 1), and all fell within a similar region of the hierarchy. As short proximal flagellum localisations bear some resemblance to a transition zone localisation, we also measured distance from the kinetoplast to the start of the mNG fluorescent signal for each proximal protein with low cell body background fluorescence, in comparison to basal body, transition zone, radial spoke and flagellum attachment zone (FAZ) proteins as controls[45–51] (Fig. 2B). This showed that signal for all measured proteins either starts at the basal plate marker Basalin[48] or more distally, so are part of the axoneme proper. However, the signals of two *T. brucei*-specific proteins start around the start of the FAZ and may be FAZ associated (Tb927.3.1200 and Tb927.11.5790).

We also measured the signal distribution for each *L. mexicana* cell line (Fig. 1C, D). Taking *T. brucei* and *L. mexicana* together, guided by the clustering analysis, the localisation of pDC1 and 2 and dDC1 and 2 are particularly characteristic: a ~50–50% proximal-distal distribution in *T. brucei* and ~20–80% proximal-distal in *L. mexicana*[1] (Fig. 1A–D, near-exclusively in clusters P1, D1 and D2 in Fig. 2A). From this combined evidence, we identified 2 conserved proteins as confidently having a pDC-like proximal localisation (1 known, FLAM6[42], and 1 novel, Tb927.6.1660) (Fig. 1A, C) and 4 as having a dDC-like distal localisation (1 known, LC4-like[1], and 2 novel, Tb927.8.8000, Tb927.4.4370,

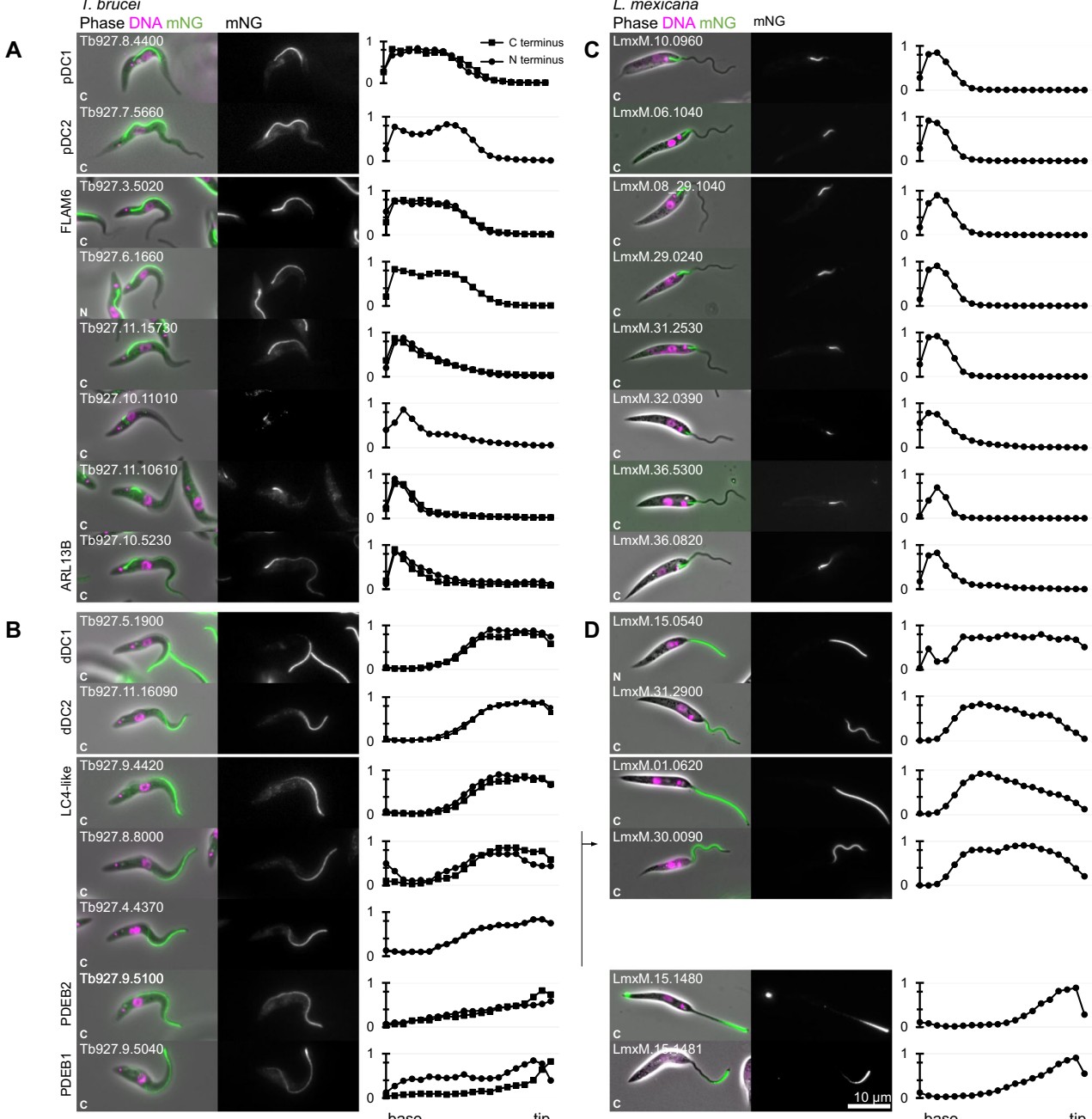

**Fig. 1 | Comparative analysis of *T. brucei* proximal and distal-specific axoneme proteins in *L. Mexicana*.revealed multiple localisation groups.**
**A**−**D** Quantitative analysis of fluorescence signal distribution along the axoneme from proximal and distal-specific axoneme proteins endogenously tagged with mNG at the N and/or C terminus. Each row corresponds to a *T. brucei* protein and its *L. mexicana* ortholog. In the first (**A**, **B**) and third columns (**C**, **D**), are an example *T. brucei* and *L. mexicana* cell, respectively. Phase contrast (grey), DNA (Hoechst 33342, magenta) and mNG (green) overlay (left) and mNG fluorescence (right) are shown. In the second (**A**, **B**) and the fourth columns (**C**, **D**), graphs representing the mNG fluorescence signal intensity along the axoneme, from the base to tip. Proteins are grouped by proximal (**A**, **C**) and distal (**B**, **D**) localisations. Data points represent the mean of *n* = 15 axonemes in 1K1N cells, normalised by maximum signal intensity per cell. Source data are provided in the Source Data file.

Fig. 1B, D). In *L. mexicana* the novel proteins are a paralogous pair: LmxM.29.0240 (proximal) and LmxM.30.0090 (distal), a duplication presents in the common ancestor of kinetoplastids like for pDC1 and dDC1, and pDC2 and dDC2 (Figure S1 A-C). In *T. brucei* the novel pair are Tb927.6.1660 (proximal) and Tb927.8.8000 and Tb927.4.4370 (distal), where the latter is a more recent duplication due to the partial chromosome 4/8 duplication[52] (Fig. S1C).

Other asymmetric localisations differed from the pDC-like and dDC-like localisation patterns. One example localised to a long proximal domain, which gradually decreased towards the distal end of the flagellum in *T. brucei* (Tb927.11.15730) and a shorter proximal region in *L. mexicana* (LmxM.31.2530) (Fig. 1A, C)−similar to pDCs, but falling in a distinct cluster (cluster P5, Fig. 2A). A further three had a short ~20% proximal signal in both *T. brucei* and *L. mexicana* with differing rates signal drop off towards the distal tip (Fig. 1A, C, clusters P3-5 in Fig. 2A). This included ARL13B which had a slower decrease in signal towards the distal tip, as previously described in *T. brucei* procyclic forms[27]. Two proteins localised along the length of the flagellum from very low proximal to strong distal signal in *T. brucei*, albeit variable (clusters D2-5 in Fig. 2A). These were PDEB1 and PDEB2, replicating the previously

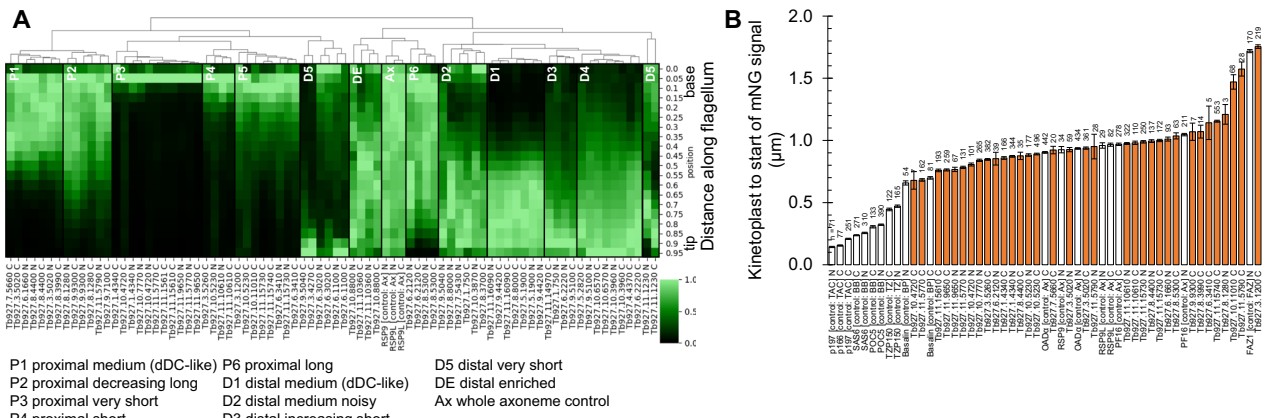

P1 proximal medium (dDC-like)   P6 proximal long              D5 distal very short
P2 proximal decreasing long     D1 distal medium (dDC-like)   DE distal enriched
P3 proximal very short          D2 distal medium noisy        Ax whole axoneme control
P4 proximal short               D3 distal increasing short
P5 proximal decreasing short    D4 distal increasing

**Fig. 2 | Quantitative analysis of *T. brucei* proximal and distal-specific sub-axonemal protein localisation. A** Hierarchical clustering of signal intensity profiles for all *T. brucei* asymmetrically distributed proteins. The heatmap represents the signal intensity profiles in (**A**, **B**) and Figs. S1, S2 and S3 from *n* = 15 axonemes in 1K1N cell. The dendrogram shows hierarchical clustering by Euclidean distance. The full-length axoneme controls RSP9 and RSP9-L are labelled Ax. **B** Distance between the kinetoplast (mitochondrial DNA) and the start of mNG-tagged protein signal for proximal axoneme proteins in *T. brucei*. Basal body (BB), transition zone (TZ), axoneme (Ax) and flagellum attachment zone (FAZ) protein controls are shown with white bars, proximal-specific proteins with orange bars. Bars and error bars represent the mean and standard error, n number of cells measured are indicated for each protein. Source data are provided in the Source Data file.

described localisation in *T. brucei* bloodstream forms[30], and we observed a corresponding localisation to the distal ~30% of the *L. mexicana* flagellum (Fig. 1B, D).

Alphafold2 structure predictions show that, of these 15 proteins, only pDC1, dDC1, pDC2 and dDC2 are extensively alpha helical (Fig. S1D). The remainder has globular domains, with at least one domain detectable based on sequence (PFAM, Superfamily, etc.)[53–55] (Fig. S1D) for each protein with the exception of LmxM.29.0240, LmxM.30.0090 and LmxM.32.0390. Comparison of pDC1 with dDC1, pDC2 with dDC2 and LmxM.29.0240 with LmxM.30.0090 all show clear similarities in predicted structure of the proximal and distal-specific protein (Fig. S1E).

The complete list of asymmetrically distributed proteins in *T. brucei* and orthologs in *L. mexicana* is summarised in Supplementary Data 1. As protein tagging generates a mutant fusion protein it is possible some of these asymmetries are aberrant tagged protein behaviour, however observing a similar localisation by N and C terminal tagging in *T. brucei* and an analogous localisation in *L. mexicana* gives confidence in their veracity. Together, these data suggest at least two (DC-dependent and independent), if not more, types of proximal and distal asymmetries.

### Deletion of asymmetrically distributed proteins has little impact on axonemal structure

Deletion of axoneme proteins can cause disruption of axonemal structure, therefore we generated deletions of each protein with proximal or distal-specific localisations in *L. mexicana*. Deletion of the respective open reading frames (ORFs) was confirmed using diagnostic PCR from purified genomic DNA from each cell line (Fig. S5). We were able to generate all deletion mutants by replacement of both alleles with drug-selectable markers, indicating that none of these proteins are vital. Flagellum length was near normal in all except one deletion mutant: ΔARL13B had shorter flagella (Fig. 3A, B) as previously observed in *T. brucei*[20] along with many other species[20,56]. This confirms ARL13B is functioning as expected in *Leishmania*, consistent with ARL13B mutations in humans causing Joubert syndrome ciliopathy[26,57,58]. All but one of the proteins are therefore more likely to be axoneme cytoskeleton components rather than factors necessary for axoneme assembly.

To identify any critical role in forming the axoneme ultrastructure, we analysed the flagellum structure of the deletion mutants by thin section transmission electron microscopy. In trypanosomatids, the flagellum protrudes from an invagination at the flagellar base – the flagellar pocket. The short pDC axoneme region of *L. mexicana* means that all cross-sections of flagella protruding through a flagellar pocket will correspond to the region containing proximal-specific proteins (Fig. 3C, E), while cross-sections through free flagella will vary: most (~90%) will be in the dDC-containing axoneme region (Fig. 3D, E), but the proportion of PDEB-containing axoneme regions will be lower. No large defects in the axoneme organisation were visible: the ninefold symmetry of the outer doublets and the central pair was unchanged (Fig. 3C–E first columns). As might be expected for proteins restricted to a small axoneme sub-domain, no proteins were necessary for normal overall axoneme organisation.

Next, we analysed changes in electron density in the flagellar-pocket and free-flagellum axoneme cross sections in these deletion mutants to identify subtle changes to axoneme structure. Making use of the ninefold radial symmetry of the outer dynein arms, we first perspective corrected then ninefold rotationally averaged the axoneme cross-section images[59,60]. Subsequently, we used ~25 rotationally averaged axoneme images to generate average flagellar pocket and free flagellum axoneme electron density maps (Fig. 3C–E second columns) and performed statistical comparisons to detect changes in electron density from the parental cell line (Fig. 3C–E third and fourth columns). We confirmed the validity of this methodology by applying it to ΔpDC1, ΔpDC2, ΔdDC1 and ΔdDC2. Both dDCs are necessary for distal (but not proximal) ODA assembly. Contrastingly, neither pDC are necessary for proximal ODA assembly, as dDCs relocalise to fill the entire axoneme[1]. Concordantly, ΔpDC1 and ΔpDC2 had subtle changes in electron density in the ODAs and at the point of attachment of the ODAs to the outer doublets in flagellar-pocket flagellar cross-sections (Fig. 3C), while ΔdDC1 and ΔdDC2 had a large loss of electron density in the ODAs in free-flagellum flagellar cross-sections (Fig. 3D).

We predicted that among proximal or distal proteins with a pDC or dDC-like localisation will be associated with the ODAs, and any electron density change in the deletion mutant will be in the DC/ODAs. In flagellar-pocket axoneme cross-sections, ΔLmxM.29.0240 had a small loss of electron density at the base of the ODAs (Fig. 3C) and both ΔFLAM6[42,61] and ΔLmxM.31.2530 (both predicted to have a cyclic nucleotide-binding domain) had a clear loss of electron density within the ODAs. All three proteins had localisations similar to pDC1 and pDC2 in *T. brucei* and *L. mexicana*.

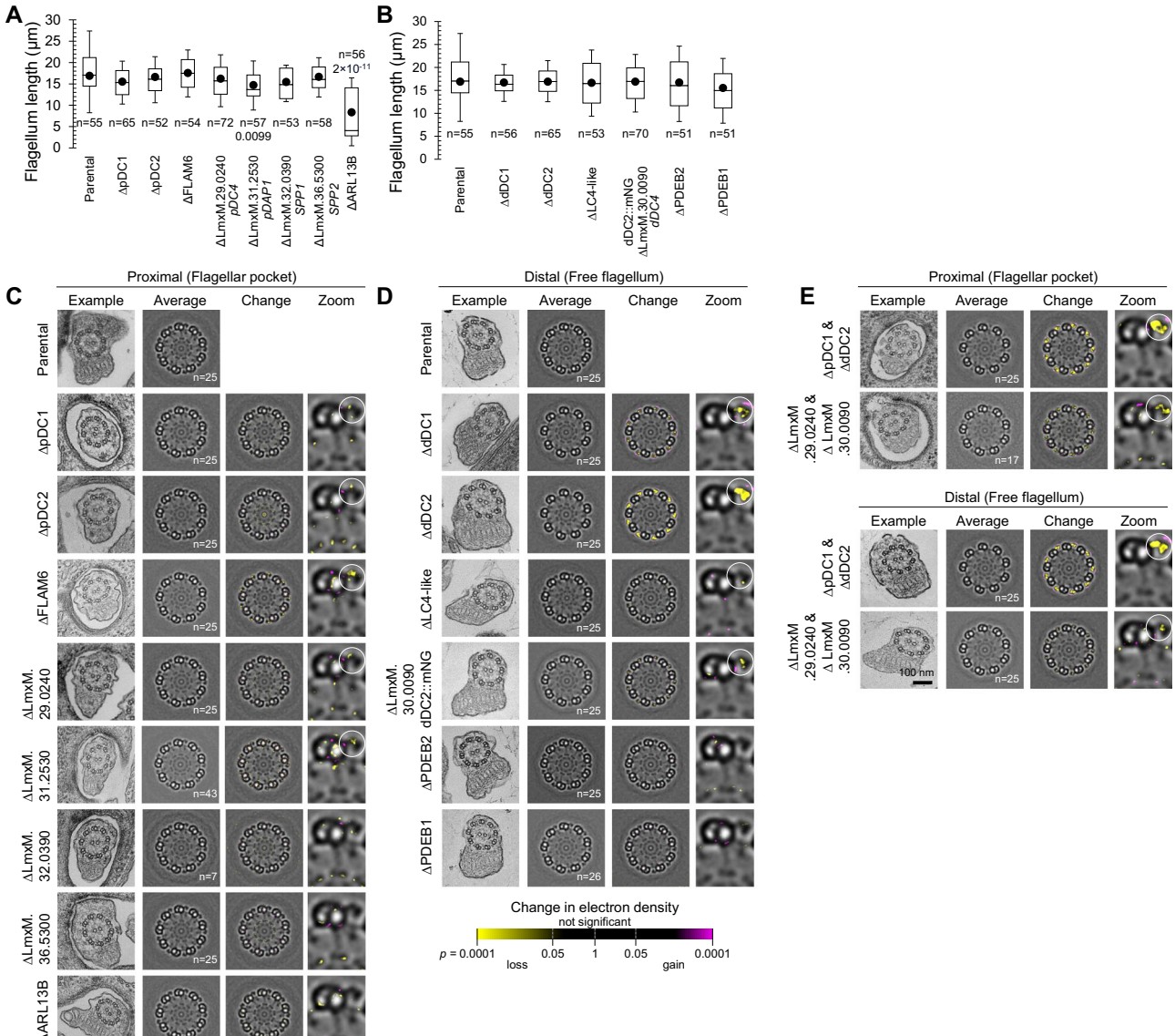

**Fig. 3 | Deletion of proteins with pDC and dDC-like localisations cause minor changes in axonemal structure. A**, **B**. Box and whisker plot of flagellum length in deletion mutants. Points represent the median; box and whiskers represent the quartile ranges and the 5th and 95th percentile. n indicates the number of cells. *P*-values were calculated using a Student's *t*-test (two tailed), *p* values <0.05 are shown. Source data are provided in the Source Data file. **C–E** Thin-section electron microscopy of axoneme structure. The first column of each shows one representative axoneme cross-section in the flagellar pocket or in the free flagellum; the second column shows an averaged axoneme structure, in which axoneme cross-

sections have had perspective deviation from circularity corrected (*n* indicates the number of axonemes used); the third column shows the result of ninefold rotational averaging and averaging across multiple axonemes; the fourth column shows a zoomed view of one microtubule doublet. **C** Analysis of the proximal flagellum for deletion mutants of proximal axoneme proteins. **D** Distal flagellum for deletion mutants of distal axoneme proteins. **E** Proximal and distal flagellum for double deletion mutant of the proximal and distal paralogous pair LmxM.29.0240 and LmxM.30.0090 in comparison to a pDC1 and dDC2 double deletion mutant. Circles in **C–E** highlight changes in electron density.

Contrastingly, deletion mutants of proteins with a short proximal localisation dissimilar to pDC1 and pDC2 (*Δ*LmxM.32.0390, *Δ*LmxM.36.5300, *Δ*ARL13B) had small or undetectable change in ODA electron density, and no clear changes elsewhere (Fig. 3C). These data were, however, relatively noisy, with changes in electron density around the central microtubule doublet and loss of electron density at the radial spoke head in several deletion mutants. As these scattered electron density changes were also observed in *Δ*pDC2 (Fig. 3C), they appear to be spurious, perhaps due to small variability in doublet position. However, they may reflect a true, but complex, biological change.

Similarly, for distal proteins with localisation similar to dDC1 and dDC2, in free-flagellum axoneme cross-sections, both *Δ*LC4-like and *Δ*LmxM.30.0090 had loss of electron density in the ODAs (Fig. 3D). For

the distal enriched PDEs, neither showed any detectable change (Fig. 3D). Overall, the most significant changes in axoneme electron density were ODA-associated and for deletion mutants of proteins with localisation similar to pDC or dDC. This suggests ODAs possess the primary structural asymmetries, although our method is only sensitive to proximal-distal, rather than doublet-to-doublet, asymmetries.

As LmxM.29.0240 and LmxM.30.0090 are a paralogous pair, we asked whether a deletion mutant of both would have a cumulative defect, with loss of ODA electron density through the entire axoneme. We performed this analysis in comparison to the deletion of both pDC1 and dDC2, which causes complete loss of ODAs (Fig. 3E). As predicted, both flagellar pocket and free flagellum flagellar cross sections showed loss of ODA electron density, albeit less pronounced than when pDC1 and dDC2 were deleted (Fig. 3E).

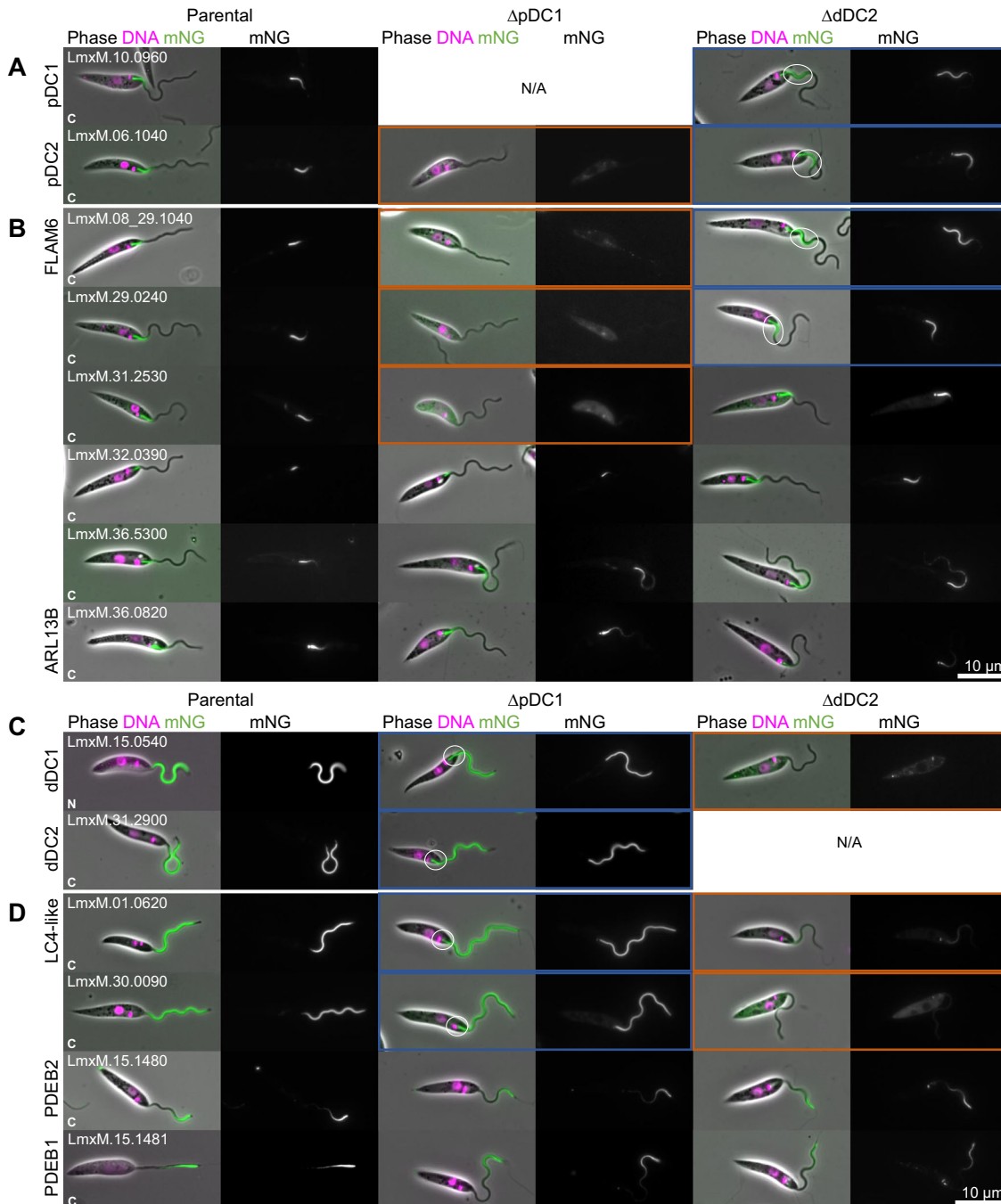

**Fig. 4 | Proximal and distal-specific localisations can be either DC asymmetry dependent or independent.** Protein localisation changes on pDC1 or dDC2 deletion. In the first column micrographs of *L. mexicana* cell line expressing tagged proximal and distal proteins. Second column: after deletion of both alleles of pDC1. Third column: after deletion of dDC2. Phase contrast (grey), DNA (Hoeschst 33342, magenta) and mNG (green) overlay and mNG fluorescence are shown. Tagged proteins are grouped by being a DC1 or 2 ortholog and by proximal or distal localisation. **A** pDC1 and 2. **B** Other proximal localising proteins. **C** dDC1 and 2. **D** Other distal localising proteins. Cell lines where the protein fails to localise to the axoneme are outlined in orange, and cell lines where the localisation is axonemal but changed are outlined in blue. Increased length of proximal signal are highlighted with a circle.

## There are multiple mechanisms for proximal and distal localisations

To test whether proximal or distal-specific proteins are dependent on the DC asymmetry, we tested whether their localisation is changes on deletion of DC proteins. First, for proteins with a proximal localisation, we deleted pDC1 in the respective cell lines expressing the tagged protein, and the successful deletion was confirmed by diagnostic PCRs (Figure S6). As previously observed in *T. brucei*[1], this causes loss of pDC2::mNG proximal flagellum signal, as pDC1 and pDC2 are co-

dependent for their normal localisation (Fig. 4A). ΔpDC1 also caused loss of proximal axoneme signal for all proteins normally with a localisation similar to pDC1 or pDC2 (FLAM6::mNG, LmxM.29.0240::mNG and LmxM.31.2530::mNG) (Fig. 4B). None of the remaining proteins that normally show a proximal localisation (LmxM.32.0390::mNG, LmxM.36.5300::mNG and ARL13B) were lost from the proximal axoneme signal on pDC1 deletion (Fig. 4B). To confirm this result, we deleted dDC2 in these tagged lines. Again, as previously observed, this causes extension of pDC1::mNG and pDC2::mNG signal to fill ~30% of the flagellum

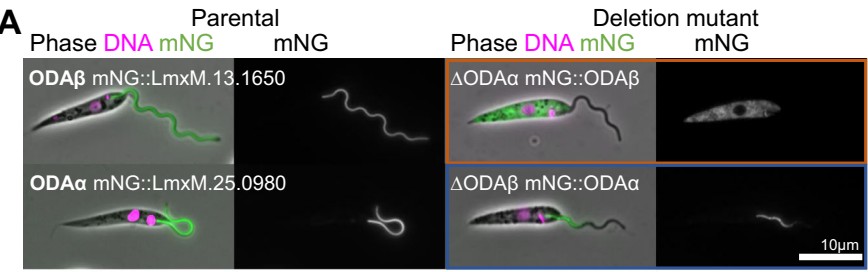

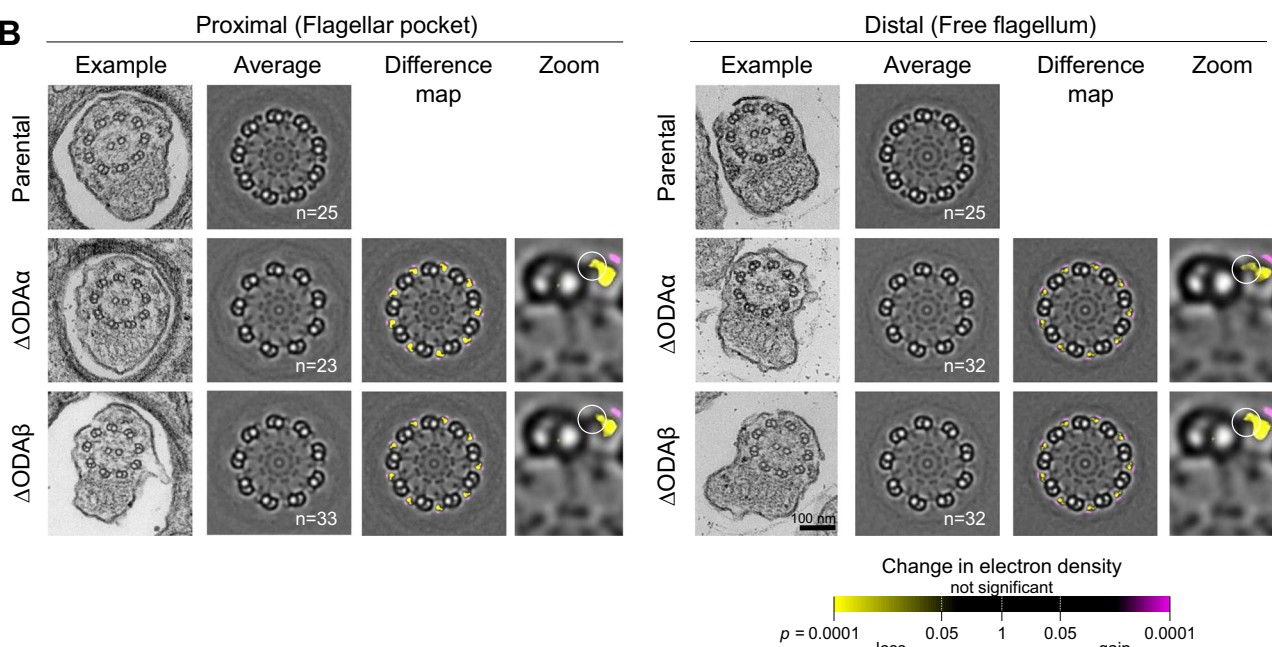

**Fig. 5 | ODAβ deletion limits ODAα incorporation in the proximal axoneme while ODAβ requires ODAα for axonemal incorporation. A** First column: micrographs of *L. mexicana* cell line expressing ODAα or ODAβ tagged with mNG at the C terminus. Second column: after deletion of both alleles of ODAα or ODAβ. Phase contrast (grey), DNA (Hoeschst 33342, magenta) and mNG (green) overlay and mNG fluorescence are shown. Cell lines where the protein fails to localise to the axoneme are outlined in orange, and cell lines where the localisation is axonemal but changed are outlined in blue. **B** Ultrastructure changes upon ODAα or ODAβ-deletion. First and fifth columns: one representative axoneme cross-section in the flagellar pocket and in the free flagella. Second and sixth columns: an averaged axoneme structure. (*n* indicates the number of axonemes used). Third and seventh columns: an electron density difference map, resulting from subtraction of deletion mutant average axoneme image from the parental cell line. Yellow indicates a loss of electron density in the deletion mutant, magenta indicates gain.

(Fig. 4A)[1]. All three proteins with a pDC-like localisation behaved similarly upon dDC2 deletion while the remainder were unaffected (Fig. 4B).

We next carried out the corresponding experiment for proteins with a distal localisation. Here, dDC2 deletion causes loss of distal axoneme signal for dDC1::mNG and LC4-like::mNG (Fig. 4C, D), as would be predicted from our previous work[1], and LmxM.30.0090::mNG behaved similarly (Fig. 4D). However, PDEB1::mNG and PDEB2::mNG were unaffected (Fig. 4D). pDC1 deletion causes expansion of the signal of dDC1::mNG, dDC2::mNG and LC4-like::mNG to fill the entire axoneme, again as expected[1] (Fig. 4C, D), and LmxM.30.0090::mNG behaves similarly (Fig. 4D). PDEB1::mNG and PDEB2::mNG localisation were unaffected (Fig. 4D).

We have therefore identified a cohort of conserved proximal (FLAM6, LmxM.29.0240 and LmxM.31.2530) and distal (LC4-like and LmxM.30.0090) proteins dependent on the DC asymmetry. As not all proximal and distal localisations are DC dependent, this confirms that there are likely multiple mechanisms for proximal and distal-specific localisation occurring in trypanosomatid flagella.

**A subset of DC-dependent asymmetrically positioned proteins are ODA heavy chain associated**

To gain insight into where proteins with a DC-dependent localisation may sit within the ODA-DC complex, we tested whether their localisation was dependent on ODA dynein heavy chains. To do so, we first tested the co-dependency of ODA dynein heavy chains on their localisation. Similar to humans, trypanosomatids have two ODA dynein heavy chains, ODAα and ODAβ. These are orthologous to *Chlamydomonas reinhardtii* ODAγ and ODAβ, respectively. In *C. renhardtii*, the ODA complex assembles in the cytoplasm prior to transport into the axoneme[62], and mutation of either *C. reinhardtii* ODAγ or ODAβ can completely prevent ODA assembly[63]. However, the precise phenotype depends whether the heavy chain is completely disrupted, like the ODAβ mutant *oda4*, which lack ODAs[64], or is truncated, like the ODAβ truncation mutant *oda4-s7*, which does not[65]. Overall, we expect deletion of either ODAα or ODAβ in *L. mexicana* to prevent ODA assembly. As expected, deletion of ODAα in a cell line expressing ODAβ::mNG showed complete loss of ODAβ::mNG signal from the axoneme. However, surprisingly, deletion of ODAβ in a cell line expressing ODAα::mNG caused loss of ODAα::mNG signal only from the distal axoneme (Fig. 5A, confirmation of deletion in Fig. S7A.).

We confirmed this result by investigating the *Δ*ODAα and *Δ*ODAβ cell lines by thin section electron microscopy, investigating proximal (flagellar-pocket) and distal (free-flagellum) cross-sections in both deletion mutants (deletion validation is shown in Figure S7B). In *Δ*ODAα, ODAs were completely absent, while ODAs were only completely absent in the distal flagellum in *Δ*ODAβ. In the proximal flagellum (sections in

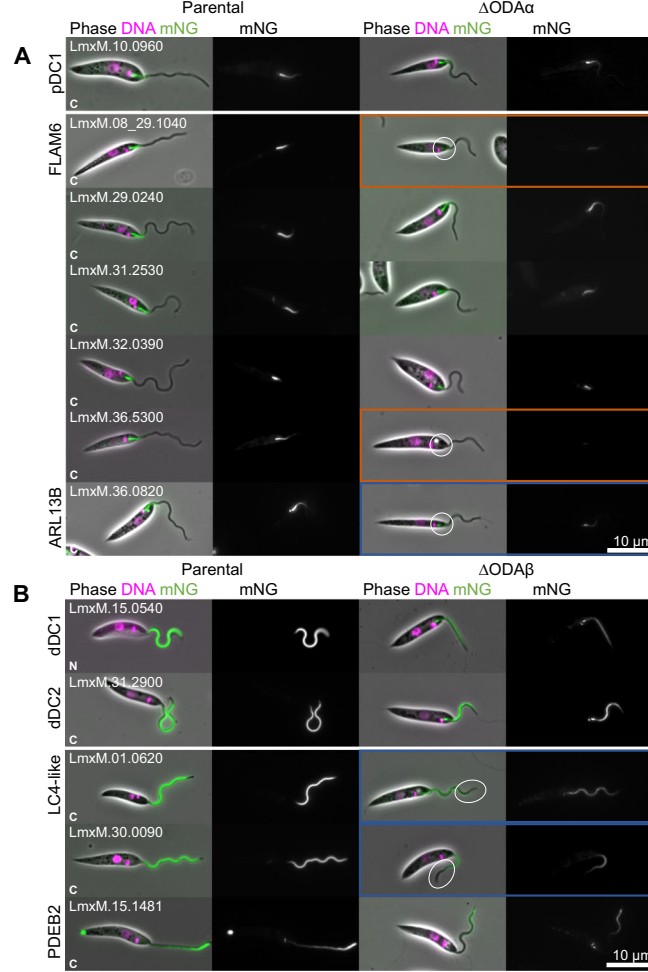

**Fig. 6 | Some non-DC-dependent localisations are ODA dynein heavy chain dependent and some DC-dependent localisations are ODA heavy chain independent. A** Micrographs of *L. mexicana* cell line expressing proximal proteins before and after deletion of both alleles of ODAα and **B** distal proteins tagged with mNG at the C terminus, before and after deletion of both alleles of ODAβ. Cell lines where the protein fails to localise to the axoneme are outlined in orange, and cell lines where the localisation is axonemal but changed are outlined in blue.

the flagellar pocket), part of the ODA was present – it appeared that the innermost half of the ODA remained (Fig. 5B). This is consistent with the expected position of ODAα as the innermost ODA dynein heavy chain, but not with ODA assembly requiring full cytoplasmic pre-assembly prior to flagellum import. Therefore, ODAs require ODAα but, unlike in *Chlamydomonas*, ODAs can incompletely assemble preferentially in the proximal axoneme in the absence of ΔODAβ.

On the basis of this result, we analysed whether proximal or distal proteins depend on the ODA heavy chains for their localisation. For proximal proteins, this required deletion of ODAα (Fig. 6A, validation of deletion Fig. S8A), while for distal proteins we deleted ODAβ (Fig. 6B, validation of deletion Fig. S8B). As expected, all DCs tested (pDC::mNG, mNG::dDC1 and dDC2::mNG) are not dependent on an ODA heavy chain for their normal localisation (Fig. 6A, B). Among proximal proteins, FLAM6::mNG, LmxM.36.5300::mNG, ARL13B::mNG signal was lost on ODA heavy chain deletion (Fig. 6A). Among distal proteins, LC4-like::mNG and LmxM.30.0090::mNG localisation were altered on ODA heavy chain deletion, both with reduced signal and failure to localise all the way to the flagellar tip (Fig. 6B). Normal distal PDEB2::mNG signal was ODA-independent. This reveals complexity: proteins can be proximal-specific and associated with the ODAs, with their proximal localisation either dependent (FLAM6, LmxM.36.5300)

or independent (LmxM.36.0830) on the asymmetry of the DCs attaching the ODAs to the doublet (Fig. 5B), suggestive of two different mechanisms proximal-distal asymmetry of ODA-associated proteins.

## LmxM.29.0240 is necessary for pDC assembly

The pDC-dependent and ODA heavy chain-independent localisation of LmxM.29.0240 and the necessity of LmxM.29.0240 and its distal paralog LmxM.30.0090 for normal ODA assembly suggests that these are additional DC components. To dissect which proteins are necessary for the DC assembly, we first deleted proteins which were dependent on the pDC for their normal proximal localisation in cells expressing pDC1::mNG localisation or deleting proteins with a distal localisation which were dependent on the dDC in cells expressing dDC2::mNG (validation of cell lines in Figure S9). LmxM.29.0240 deletion affected normal pDC1::mNG localisation, giving a shorter proximal signal and accumulation around the basal body, while deletion of FLAM6 or LmxM.31.2530 did not affect pDC1::mNG localisation (Fig. 7A, validation of cell lines in Fig. S9A). LC4-like or LmxM.30.0090 deletion did not affect the distal localisation of dDC2::mNG, although some accumulation of dDC2::mNG near the flagellum base occurred following LmxM.30.0090 deletion (Fig. 7B, validation of cell lines in Fig. S9B). LmxM.29.0240 is, therefore, acting as a pDC component necessary for pDC assembly; thus, we named it pDC4 (skipping DC3, as it appears unrelated to *C. reinhardtii* DC3[66,67]). Following our naming scheme for the proximal and distal DC1 and DC2 paralogous pairs, we named LmxM.30.0090 dDC4—although it was not necessary for dDC assembly it was necessary for normal ODA assembly (Fig. 2E).

To comprehensively test whether pDC4 is truly necessary for pDC assembly and map the co-dependencies of pDC-associated components, we used combinatorial tagging and deletion. pDC4 was necessary for the normal localisation of FLAM6::mNG and LmxM.31.2530::mNG in addition to pDC1::mNG (Fig. 7C, validation of cell lines in Figure S9C). Normal LmxM.31.2530::mNG localisation was also dependent on FLAM6. pDC1, pDC2 and pDC4 therefore form the pDC, with FLAM6 and LmxM.31.2530 binding to the structure, LmxM.31.2530 in a FLAM6-dependent manner.

## Proximal and distal-specific proteins contribute to the control of the tip-to-base flagellum beat

Proteins that are not vital for flagellum assembly or normal ultrastructure but are well-conserved between related species are good candidates for coordinating a normal flagellar beat. To identify any correlation of beat regulation function with proximal or distal asymmetry, we carried out detailed cell swimming and flagellum beat analysis of all proximal and distal proteins not necessary for normal flagellum assembly. We carried out three analyses (Fig. 8, S11): (1) Average cell swimming speed, (2) proportion of cells undergoing different types of flagellar beat (high-frequency tip-to-base symmetric, flagellar-type; low-frequency base-to-tip asymmetric, ciliary-type; low-frequency aperiodic movement, static/uncoordinated) and (3) beat waveform properties (where amplitude, frequency, and number of waves per flagellum tended to change, Fig. S11). This multi-scale analysis aims to identify beat changes which lead to swimming changes, however there may be discrepancies. Different assays were carried out under different physical conditions (1, in a deep volume and analysed away from the slide and coverslip to avoid surface interaction effects[16]; 2 & 3, in a thin volume between a slide and coverslip to keep the cells in focus), exposed the cells to different lighting and heating (due to different microscope magnifications: 1, ×10; 2, ×20; 3, ×63), and had different observation biases (1 & 2, relatively unbiased; 3, only cells undergoing a periodic tip-to-base waveform which stay sufficiently in focus).

First, we consider proteins with a dDC-like localisation, where our previous analyses are more comprehensive[1]. As previously described, dDC1 and dDC2 deletion caused reduced swimming speed and

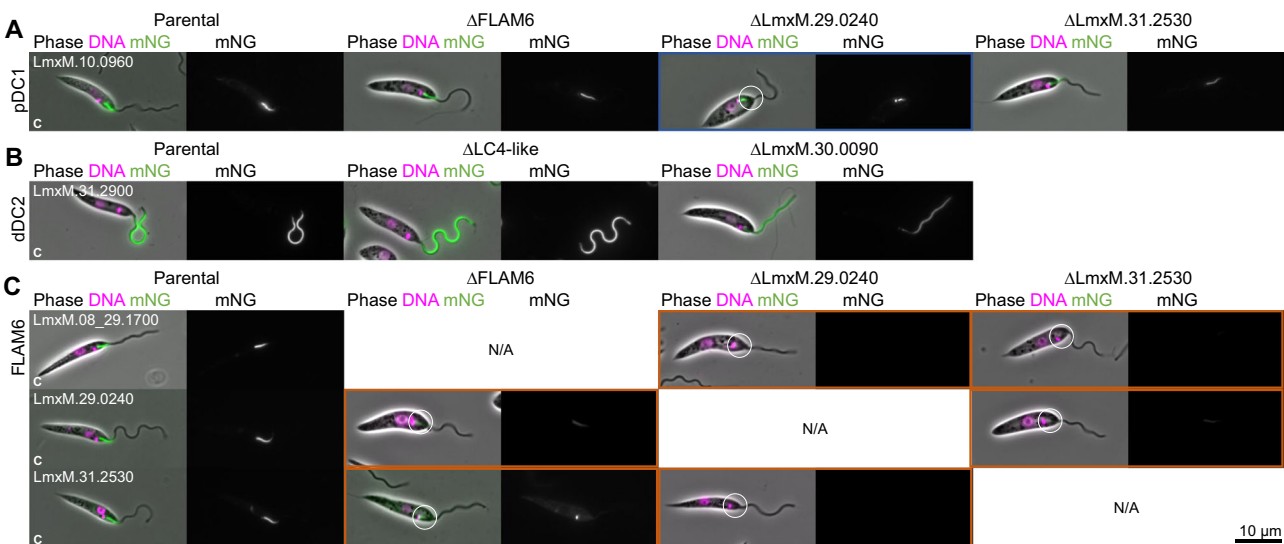

**Fig. 7 | Localisation co-dependency among DC-dependent proteins identifies an additional DC component. A** Change in pDC localisation following deletion of proteins with a localisation dependent on pDC. In the first column, micrographs of *L. mexicana* cell line expressing proteins tagged with mNG at the C terminus. In columns 2 to 4, micrographs after deletion of FLAM6, LmxM.29.0240 or LmxM.31.2530. **B** Change in dDC localisation following deletion of proteins with a localisation dependent on dDC. Micrographs of *L. mexicana* cell line expressing

dDC2 protein tagged with mNG at the C terminus, before and after deletion of both alleles of LC4-like and LmxM.30.0090. **C** Combinatorial tagging and deletion of pDC1 and proteins analysed in A. In the first column, micrographs of *L. mexicana* cell line expressing proteins tagged with mNG at the C terminus. In columns 2 to 4, micrographs after deletion of FLAM6, LmxM.29.0240 or LmxM.31.2530. Cell lines where the protein fails to localise to the axoneme are outlined in orange, and cell lines where the localisation is axonemal but changed are outlined in blue.

reduced ability to carry out the normal tip-to-base symmetric beat (Fig. 8A, B). When they did undergo a tip-to-base symmetric beat, it was much lower frequency. Again, as previously described, LC4-like deletion caused a small but significant increase in swimming speed, arising from a higher tip-to-base symmetric beat frequency. Deletion of the DC component LmxM.30.0090 (dDC4), also reduced swimming speed and time spend undergoing a tip-to-base waveform, but not the same clear decrease in beat frequency observed for dDC1 or dDC2 deletion (Fig. 8C).

Deletion of either distal enriched PDEBs had little effect on swimming speed (Fig. 8A) and reduced the number of cells able to undergo a tip-to-base symmetric beat (Fig. 8B); however, those that did had a larger range of beat frequencies (Fig. 8C). Distal enriched PDEs are therefore potential positive regulators of tip-to-base symmetric beat frequency.

Next, we considered proteins with a proximal localisation. In our previous work[1], no large effect of pDC1 deletion on cell swimming speed was observed, thus was not analysed in detail. Here, consistent with this, pDC1, pDC2 deletion had little effect on swimming speed, and LmxM.29.0240 deletion (pDC4) was similar (Fig. 8D). However, pDC2 and LmxM.29.0240 deletion caused less tip-to-base symmetric beating, with a prominent increase in base-to-tip asymmetric beats upon LmxM.29.0240 deletion (Fig. 8E). pDC1 and pDC2 deletion changed tip-to-base symmetric beat frequency, giving a bimodal distribution of frequencies, i.e. cells tended to have either faster or slower beat frequency (Fig. 8F). As pDC1, pDC2 and LmxM.29.0240 are co-dependent for assembly of the pDC, we might expect these deletion mutants to have similar phenotypes. There was, however, some mutant-to-mutant variability, but overall pDC proteins were necessary for a normal beat profile.

Among the other proximal proteins, FLAM6, LmxM.31.2530, LmxM.31.0240 and LmxM.32.0390, deletion had little effect on swimming speed (Fig. 8D) or proportion of time spent exhibiting different beat types (Fig. 8E). However, all four had a large increase in beat frequency of tip-to-base symmetric beats (Fig. 8F). A naïve prediction is that a proximal or distal-specific protein may influence only the proximal or distal waveform respectively. However, analysing

frequency, amplitude and wavelength for the proximal and distal flagellum separately showed no significant difference between the two for any mutant with altered overall beat frequency (Figure S12). Therefore, surprisingly, the presence of proximal-specific axoneme proteins can negatively regulate the flagellum-wide beat frequency of beats starting at the far end of the flagellum.

Finally, to complete the characterisation of the paralogous pair of DC components, LmxM.30.0090 and LmxM.29.0240 (pDC4 and dDC4), we asked how swimming and beating changed on deletion of both, in comparison to complete removal of the DC by deletion of pDC1 and dDC2. Complete loss of the ODAs by deletion of pDC1 and dDC2 greatly reduced the swimming speed, occurrence of tip-to-base waves and frequency of tip-to-base beating, with an intriguing increase in beat amplitude (Fig. 8E–G). Deletion of both LmxM.30.0090 and LmxM.29.0240 had a less pronounced phenotype, with decreased swimming speed, occurrence of tip-to-base waves and low frequency tip-to-base waves in a subset of flagella (Figs. 8E and 7G).

## Discussion

We have shown that proximal-distal asymmetry of axonemal organisation is a complex phenomenon in highly motile trypanosomatid parasites, identifying 51 examples of flagellar proximal- or distal-specific protein localisations in *T. brucei*. However, only ~30% (16/51) were conserved, with an *L. mexicana* ortholog with an equivalent asymmetric localisation (summarised in Fig. 9). While future more sensitive analysis may detect further orthologs, any will be of very low sequence similarity. This shows trypanosomatid lineage-specific adaptations on the time scale of hundreds of millions of years. Many asymmetrically positioned *T. brucei* proteins lacked an *L. mexicana* ortholog (~50%, 26/51), with the remainder localising to whole flagellum rather than an asymmetric sub-compartment – indicating either loss of asymmetry in *L. mexicana* or gain of asymmetry in *T. brucei*.

By focusing on conserved factors in key motile life stages of these two parasites we aimed to analyse factors necessary for flagellar beating. However, it is important to recognise that trypanosomatid parasites undergo dramatic flagellum adaptation for different stages

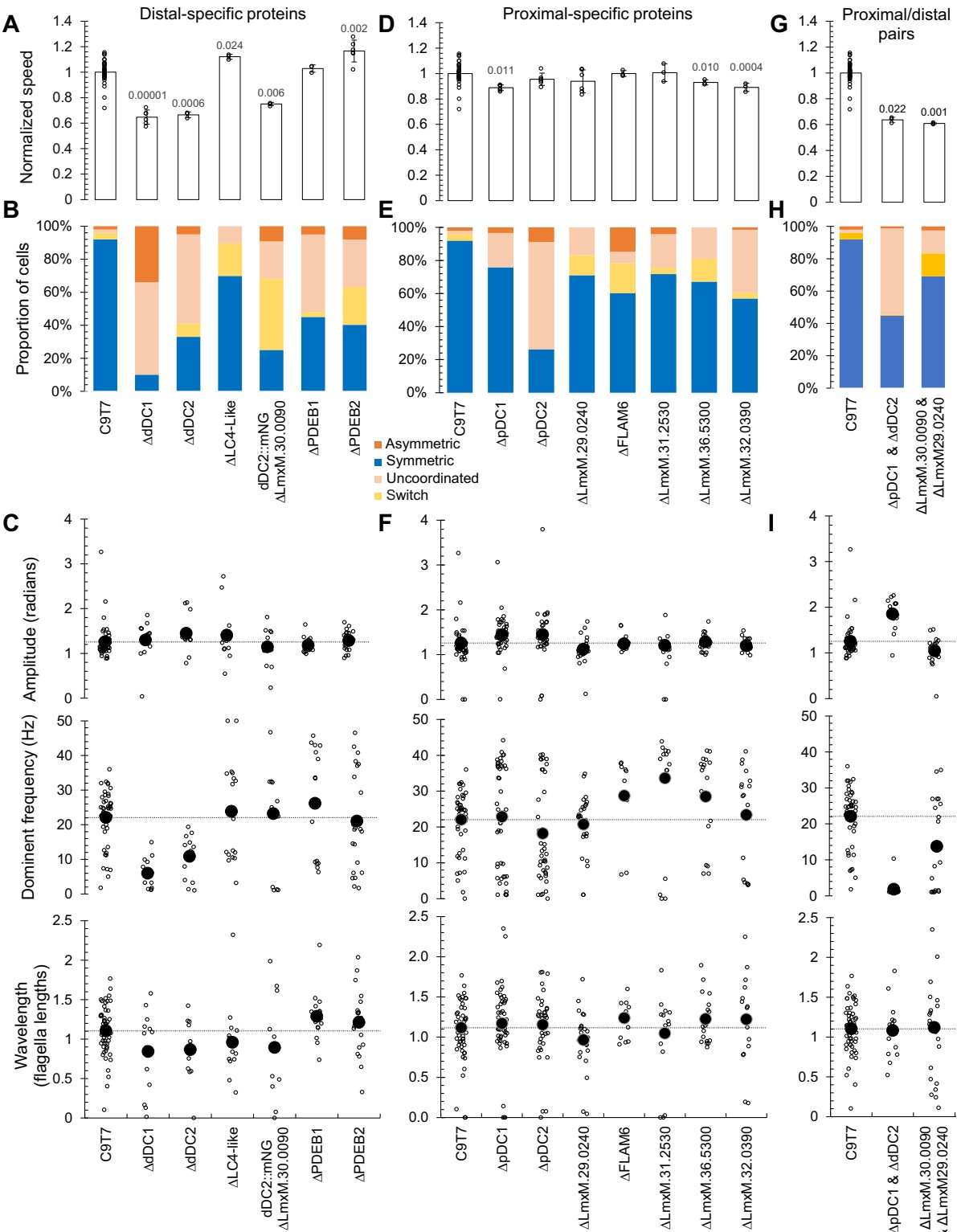

**Fig. 8 | Distal and proximal proteins contribute to the control of flagellum beat and their frequency.** Data separated into analysis of deletion mutants of distal (**A**–**C**) proximal (**D**–**F**) localising proteins and select double deletion mutants (**G**–**I**). **A**, **D**, **G**. Swimming speed normalised to parental C9T7 cell line. The data are represented as the mean ± SDs of triplicate or sextuple (parental) samples. p values from two-tailed Student's *t*-test, *p* < 0.05 are shown. **B**, **E**, **H** The proportion of cells undergoing different types of flagellar movement, comparing deletion mutants to the parental C9T7 cell line. **C**, **F**, **I** Amplitude, dominant frequency and wavelength per flagellum in parental cell line and knockout distal mutants. Open circles represent individual cells, solid circles represent the mean. The horizontal dotted line is the mean of the parental C9T7 cell line. Source data are provided in the Source Data file.

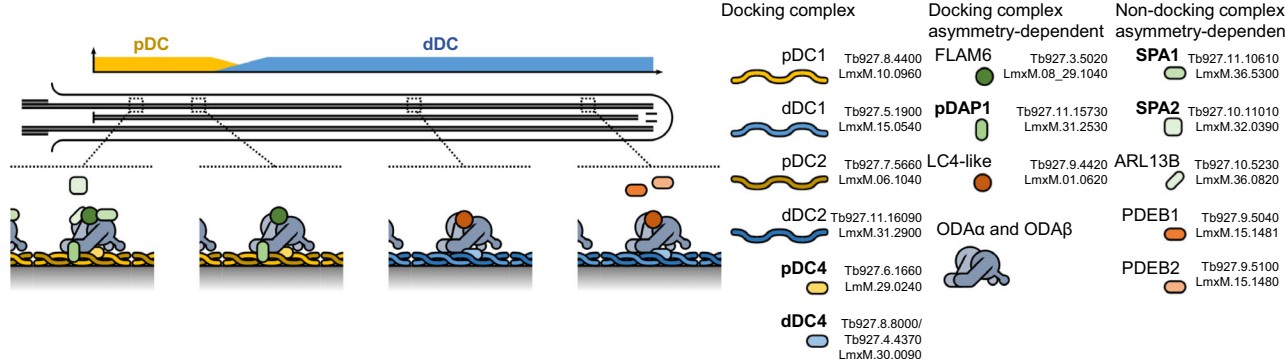

**Fig. 9 | Summary of conserved proximal-distal asymmetry in the *Leishmania* and *Trypanosoma* flagellum.** Cartoon summary of asymmetrically localised axonemal proteins, where proteins drawn overlapping summarises dependency for assembly, working out from the DC.

of the species-specific life cycles, for example for surface attachment or shortening to barely protrude from the flagellar pocket[68]. Some proteins with species-specific asymmetric distributions may be involved in these species-specific adaptations.

We dissected the conserved axoneme asymmetry, which lead to definition of the conserved proximal and distal docking complex machinery and associated proteins. This included a new paralogous pair of proteins associated with and partially necessary for assembly of the proximal and distal docking complex: LmxM.29.0240 and LmxM.30.0090 which we name pDC4 and dDC4 respectively. We also discovered that LmxM.31.2530 is a **p**roximal **D**C-**a**ssociated **p**rotein not necessary for pDC assembly (which we name pDAP1) and that FLAM6[42], like LC4-like[1], is dependent on the DC asymmetry for its asymmetric localisation. Not all asymmetric localisations were DC-dependent, and we identified proteins which localise to a **s**hort **p**roximal **a**xoneme region (LmxM.36.5300 and LmxM.32.0390 which we name SPA1 and SPA2) independent of the pDC.

These different dependencies point to different mechanisms of generation of proximal-distal asymmetry. Previously, we saw that the DC-based proximal-distal asymmetry is likely achieved by mutually exclusive binding to the axoneme in combination with intraflagellar transport (IFT) transport to form a proximal-distal concentration gradient[1], and recent work identified LRRC56 in *T. brucei* as IFT cargo necessary for correct incorporation of distal DCs[69]. Not all proximal-distal asymmetries seem arise from similar mutual exclusivity: We identified short proximal (SPA1 and SPA2) and short distal regions (PDEB1 and PDEB2), but no corresponding long proximal and long distal regions.

Short distal and short proximal regions might be explained by limiting protein abundance, with their localisation maintained by anterograde or retrograde transport by IFT, respectively. A proximal localisation could also be explained by no active transport by IFT, only diffusion up the flagellum and high axoneme binding affinity. The latter was previously proposed for ARL13B[27] and the proximal ARL13B localisation was dependent on the DC for its asymmetric localisation. In these cases, there is the possibility that fluorescent protein tagging perturbed the normal expression level or ability of the protein to access and bind the axoneme, however we generally observed the same localisation by N and C terminal tagging in *T. brucei* and a consistent localisation in *L. mexicana*.

We previously noted that ODA proximal-distal asymmetries occur across diverse eukaryotes[14–19] although are not necessarily orthologous to *T. brucei* and *L. mexicana* (i.e. not all involving pDC and dDC paralogs)[1]. Interestingly, most proteins we analysed in detail were ODA-associated (Fig. 6), hinting at a tendency for proximal-distal asymmetries to generally arise in ODAs. Most proteins we characterised are not conserved outside of the kinetoplastid lineage, although orthologs of pDC4/dDC4 and SPA2 are found in the genomes

of several flagellate/motile ciliate species, including *Chlamydomonas reinhardii* (UniProt IDs A0A2K3DVV8 and A0A2K3DZJ7 respectively).

Many of these asymmetrically positioned proteins are necessary for a normal flagellar beat, summarised in Table S1, and deletion mutants generally had complex changes. Deletion of dDC1 or dDC2 caused aberrant phenotypes unlike deletion of other asymmetrically localised proteins, with fewer symmetrical tip-to-base waveforms which occurred at a lower frequency and increased base-to-tip asymmetric beats (particularly for dDC1 deletion). dDC4 deletion did not clearly phenocopy dDC1 or dDC2 deletion, however also had a weaker impact on distal ODA assembly, so this might be expected. Other deletion mutants of distal proteins often had reduction in incidence of symmetrical tip-to-base waveforms, and typically either little change to beat frequency or a bimodal distribution with some cells undergoing an increased and some a decreased tip-to-base beat frequency. Therefore, distal ODAs are needed for efficient distal waveform initiation, and the presence of asymmetrically positioned flagellar proteins is necessary for normal beat frequency.

PDEB2 deletion was previously shown to slightly increase swimming speed in *L. mexicana*[9], while PDEB deletion was previously shown to have no effect on *T. brucei* swimming speed[30]. While we saw no significant increase in *L. mexicana*, suggesting that the PDEB swimming speed may be marginal, variable, or contingent on deleting both PDEBs, we saw that beat frequency was clearly and strongly affected by deletion of either PDEB. This defective beat control may explain the defects in *T. brucei* PDEB deletion mutant chemotaxis-like behaviour ("social motility") and inability to progress through the tse-tse fly vector[29,30], and emphasises the importance of cAMP signalling to regulate flagella beating.

Roles in regulation of tip-to-base waveform frequency control are intuitive for distal-specific proteins, like PDEB. But deletion of the either FLAM6 or pDAP1, both pDC-associated, caused a significant increase in the tip-to-base beat frequency. How can removal of protein present only in a short proximal region of the flagellum increase beat frequency by promoting initiation of waveforms at the distal end of the flagellum? We can consider two key hypotheses: Their deletion may lead to a constant signal that increases the rate of tip-to-base waveform initiation thus increasing frequency, or their deletion may allow faster transmission of individual signal that initiates individual tip-to-base waveforms in more rapid succession. The former seems unlikely – this requires proximal proteins to act as sinks for a molecule that acts as a constant negative signal. FLAM6 and pDAP1 are predicted to have cAMP binding domains, and cAMP can be a positive regulator of beating[32–34]. However, the predicted binding rather than enzymatic degradation of cAMP represents a limited sequestration rather than a sink, making their deletion unlikely to have a large effect on flagellar cAMP concentration. The latter is possible, but only through limited mechanisms. A signal from the flagellar base would have to reach the

tip in ~25 ms to initiate successive tip-to-base waveforms at 40 Hz (a speed of ~1000 μm/s). Diffusing or actively transported (*cf*. IFT train speed in trypanosomatids[70]) signalling molecules are far too slow; however, a mechanical signal like shear force on the outer doublet microtubules would be transmitted almost instantaneously. We suggest that these proximal-specific proteins act to modulate shear force generation as waveforms approach the base of the flagellum.

Overall, we have shown that there are multiple mechanisms for generating proximal-distal asymmetry in axoneme organisation, often involving the ODAs, and we have mapped the proteins whose proximal-distal asymmetry is dependent on a three-component docking complex with proximal and distal paralogous variants. Surprisingly, proximal-specific proteins can be necessary for normal beat frequency for flagellar waveforms starting at the distal end of the flagellum.

## Methods

### Parasite cell lines, maintenance and culturing
Cas9T7 *L. mexicana* derived from WHO strain MNYC/BZ/62/M379, expressing Cas9 and T7 RNA polymerase[8,9] was grown in M199 (Life Technologies) supplemented with 2.2 g/L NaHCO₃ (Sigma-Aldrich, S6014), 0.005% hemin (Sigma-Aldrich, H9039), 40 mM HEPES HCl (pH 7.4) (Merk Life Science, H3375) and 10% FCS (Merk Life Science, F9665). Selection for the T7 and Cas9 constructs was with 50 μg/mL nourseothricin sulfate (Cambridge Bioscience, Ltd) and 32 μg/mL hygromycin B (Stratech, A2515). *L. mexicana* cultures were grown at 28 °C. Culture density was maintained between $1 \times 10^5$ and $1 \times 10^7$ cells/mL for continued exponential population growth. Culture density was measured using a haemocytometer. Identity was confirmed by recent mRNA and genomic sequencing, and lack of mycoplasma contamination was confirmed by fluorescent microscopy with a Hoechst 33342 DNA stain.

### Candidate protein selection and ortholog identification
Proteins with proximal or distal flagellar localisations were identified using TrypTag *T. brucei* procyclic form subcellular protein localisation data available up to 12th March 2018[71] based on TriTrypDB version 36[72]. All proteins with a flagellum, axoneme or paraflagellar rod localisation annotation were checked, therefore omission indicates no clear proximal or distal-specific localisation. To identify orthologs of *T. brucei* genes in *L. mexicana* for analysis we used syntenic orthologs listed based in TriTrypDB (based on OrthoMCL analysis)[72] supplemented by reciprocal best BLASTP[73] search hits with $e < 10^{-5}$.

### Genetic modifications
Endogenously tagged and deletion mutant cell lines were generated by transfection of Cas9T7 with the appropriate combination of DNA encoding sgRNAs and repair constructs with flanking 30 base homology arms carrying the necessary drug-selectable marker and, for tagging, the fluorescent protein[8]. In brief:

Protospacer adjacent motif (PAM) sites for sgRNAs were taken from the database previously generated for *L. mexicana* MNYC/BZ/62/M379[8]. These were designed by searching a 45 bp window upstream of the start codon and downstream of the stop codon using the Eukaryotic Pathogen CRISPR guide RNA/DNA Design Tool[74]. Homology arms for repair constructs were taken from the database previously generated for *L. mexicana* MNYC/BZ/62/M379[8]; 30 bp sequences from 45 bases into the 5′ UTR, the start of the ORF, the end of the ORF and 45 bases into the 3′ UTR.

DNA encoding the single guide RNA (sgRNA) and a T7 polymerase promoter (sgDNA) was generated by templateless PCR using two primers, one encoding the gene-specific PAM site and T7 promoter and the second one encoding the remainder of the sgRNA[8].

To generate endogenous tagged cell lines, we used repair constructs generated by long primer PCR using the pLPOT plasmid series[1]

as template DNA. The long primers introduce 30 base flanking homology arms to a fluorophore and drug-selectable marker encoded in the template plasmid[75]. For endogenous tagging with mNG, we used pLPOT mNG Neomycin. For N terminal tagging, the sgRNA targets CRISPR cutting to just upstream of the target ORF, for C terminal tagging just downstream. The flanking homology arms are homologous to the 5′ UTR and start of the ORF for N terminal tagging, and the end of the ORF and 3′ UTR for C terminal tagging[8,75]. To generate deletion mutants with both alleles replaced by drug-selectable markers, we used repair constructs generated by long primer PCR using pT Blast, pT Puro, pT Neo or pT Phleo as templates[8] and two sgRNAs. For deletion, one sgRNA targets CRISPR cutting just upstream and just downstream of the ORF, and the flanking homology arms are homologous to the 5′ and 3′ UTR[8].

For sgDNA generation, a 50 μl PCR reaction was prepared with 2 μM each of the gene-specific and standard reverse primers, with 1 unit Expand High Fidelity polymerase in 1× HiFi reaction buffer with MgCl₂ (Merk Life Science), 3% (v/v) DMSO. The thermocycle was 30 cycles of 30 s at 94 °C, 30 s at 60 °C, and 30 s at 72 °C[8,75]. For repair construct generation, a 50 μl PCR reaction was prepared with 30 ng template plasmid (pPLOT or pT), primers (2 μM each of gene-specific forward and reverse primers), with 1 unit Expand High Fidelity polymerase with Expand™ High Fidelity PCR System1x HiFi reaction buffer with MgCl₂MgCl₂ and 3% (v/v) DMSO. The thermocycle was 5 min at 95 °C, then 35 cycles of 15 s at 94 °C, 30 s at 65 °C and 60 s at 72 °C, and a final 1 min elongation at 72 °C. 1 μL of each reaction was analysed by 1% agarose gel electrophoresis to check for the presence of the expected product.

### Electroporation and drug selection
Prior to electroporation, the necessary sgRNA and repair construct PCR reactions were mixed then precipitated with 10 μL of 3 M sodium acetate (Sigma, S8750) and 300 μL of EtOH at −80 °C for 20 min, pelleted by centrifugation (20,000 g, 30 min at 4 °C), washed once with 70% EtOH, re-pelleted then resuspended in 10 μL of water. *L. mexicana* were electroporated using the Nucleofector II (Amaxa) in 2 mm cuvettes with the X-100 programme. $1 \times 10^7$ cells were mixed with 250 μL electroporation buffer (a ratio of 1:6:3 of 1.5 mM CaCl₂ H₂O (Merk, 202940) and 200 mM Na₂HPO₄ (Merk, 71640), 70 mM NaH₂PO₄ (Merk, S5011), 15 mM KCl (Merk, P5405), 150 mM HEPES pH 7.4 (Merk, H4034)). Cells were immediately resuspended in pre-warmed M199. Successful transfectants were selected and maintained using the necessary combination of drugs selection (20 μg/mL puromycin dihydrochloride ((Puro) (Cambridge Science Ltd)), 5 μg/mL blasticidin S hydrochloride ((Bla) (Stratech, B4879)), 40 μg/mL G418 disulfate ((Neo) (Invitrogen)) and 20 μg/mL phleomycin ((Phleo) (Melford, P20200)). Drugs were added 15 h after transfection, and resistant populations took 4 to 15 days to emerge. All cell lines generated are listed in Supplementary Data 2, including necessary selection drugs. Sequences of all primers are listed in Supplementary Data 3.

### Diagnostic PCR for gene knockout validation
To verify the loss of the target ORF in drug-resistant transfectants, a diagnostic PCR was performed by amplifying a short PCR product (100–300 bp) within the ORF of the target gene. We used a positive control with genomic DNA from the parental cell line to confirm successful detection of the target ORF. If no product was detected then the annealing temperature was lowered by up to 2 degrees, until a product was obtained. Primers to amplify a short fragment of the PF16 (LmxM.20.1400) ORF was amplified as a technical control to confirm the presence of genomic DNA from the knockout cell line. The PCR mixture for each reaction was: 25 to 100 ng of gDNA and 10 μM (1.25 μL) each of the forward and reverse primers mixed with the FastGene Optima HotStart Ready Mix with dye (12.5 μL) (Nippon Genetics, [P8-0082]) and made up to 25 μL with water. The

thermocycle was 3 min at 95 °C, then 35 cycles of 15 s at 95 °C, 15 s at 58 °C and 30 s at 72 °C, and a final 1 min elongation at 72 °C.

## Fluorescence microscopy

*L. mexicana* expressing fluorescent fusion proteins were imaged live. Cells were washed three times by centrifugation at 800 × g followed by resuspension in PBS. DNA was stained by including 10 μg/mL Hoechst 33342 in the second washing. Washed cells were settled on glass slides and were observed immediately. Widefield epifluorescence and phase-contrast images were captured using a Zeiss Axioimager.Z2 microscope with a 63×/1.40 numerical aperture (NA) oil immersion objective and a Hamamatsu ORCA-Flash4.0 camera.

Cell morphology and fluorescence distribution measurements were made using measurement tools in ImageJ[76]. To measure fluorescence signal distribution along the length of flagella from epifluorescence micrographs, the flagellum was manually traced in cells with one flagellum, starting at the kinetoplast (visible in the Hoechst 33342 image) then using the phase-contrast image as a reference to trace it to the distal tip. Signal intensity was measured in the background-subtracted green fluorescence channel using the "Plot Profile" tool and normalised per cell to a 0–1 range. As flagella have length variation, data were plotted on a normalised 0 to 1 distance along the flagella, and the average of 15 flagella is shown. Hierarchical clustering was carried out using the seaborn Python module using the average method and Euclidean distance metric.

Distance from the kinetoplast to the start of mNG signal was measured in Python using skimage, assisted by the tryptag Python module[77]. For each cell in each cell line listed as a 1K1N by the tryptag module, maximum pixel value in circles of increasing radius around the kinetoplast centroid (as provided by the tryptag module) was recorded and fitted to a sigmoid curve. The position of the midpoint of the sigmoid fit was taken as an individual measurement of kinetoplast to start of flagellar fluorescent signal if 1) the fitted midpoint was within the distance range measured, the slope was sharp (>0.7, based on the point spread function), the maximum of the sigmoid was high (>1000, based on typical signal intensity) and the R-squared was high (>0.95).

## Motility assays

For motility analysis in *L. mexicana*, swimming behaviours were analysed for cells in the exponential growth phase in normal culture medium using the approach we previously developed[6]: A 25.6 s video at five frames/s under darkfield illumination was captured from 5 μL of cell culture in a 250 μm deep chamber using a Zeiss Axioimager.Z2 microscope with a 10×/0.3 NA objective and a ORCA-Flash 4.0 (Hamamatsu) camera. Particle tracks were traced automatically, and mean cell speed, mean cell velocity and cell directionality (the ratio of displacement achieved to distance travelled) were calculated using our previously published software tools[9,78].

## Flagellum beat type analysis

To determine the proportion of cells in a population undergoing different beat types, 1 mL of exponential growth cells (between $1 \times 10^6$ and $1 \times 10^7$ cells/mL) was centrifuged for 5 min at 800 × g. Between 700 and 950 μL (depending on cell density) of supernatant was removed and cells were resuspended in M199 (300–500 μL). One microlitre of 5 μm polystyrene beads (Sigma 79633) diluted 1:100 in M199 was added, which ensure a 5 μm sample depth. 1 μL of cell sample was added to the centre of a 2 by 5 cm area marked with a hydrophobic pen on a slide, and a glass coverslip (1.0 thickness) added. Videomicrographs of swimming cells under phase-contrast illumination were captured with an Andor Neo 5.5 camera at 200 frames/s for 0.5 s, using 20×/0.3 NA objective lens on a Zeiss Axioimager.Z2 inverted microscope. Cells with only one observable flagellum (non-dividing) were manually classified into symmetrical tip-to-base (continuous or interrupted),

asymmetric base-to-tip, wave type switch and static and uncoordinated.

## Flagellar beating analysis

Parental cell line and deletion cell lines were analysed by high-speed video microscopy. A 5 s video at 200 frames/s under phase-contrast illumination was captured from a thin film of cell culture between a slide and coverslip using ZeissAxioimager.Z2 microscope with a 100×/1.4 NA objective and an Andor Neo 5.5 camera. Flagellar beat behaviours for each cell lines were classified manually, and only symmetrical tip-to-base waveforms were analysed for this study. Automated image analysis and flagellum tracking[79] in ImageJ (version 1.52a) was used to digitise the flagellar waveforms of a target 25 cells (all at least 19), with a target of 950 (all at least 30 frames) per cell, manually excluding cells which swam out of focus or out of the frame during video capture. Digitised waveforms were screened based on the variation in measured flagellum length over each video (a proxy for consistency of digitisation), then smoothed in space and time with smoothing splines in MATLAB. Finally, waveforms were excluded if they were poorly approximated by a sinusoidal beat (wild-type *L. mexicana* has a sinusoidal tip-to-base beat), as measured by a least-squares fit. A range of beating characteristics were computed for each cell and all deletion cell lines were compared to the parental cell line: Angular amplitude is defined in terms of the angle of the tangent to the flagellum: for every tracked point along the flagellum, we measure the angle the tangent makes with the cell body and record the range of values taken over a beat cycle. The overall angular amplitude is then the maximum of these ranges along the flagellum. The beat frequency is computed via a Fourier transform of the tangent-angle representation of the beat, aggregated along the length of the flagellum. We report the single frequency that dominates the power spectrum, the dominant frequency, which is a well-defined measure as *Leishmania* beating is almost a pure single frequency[80]. All code is available at Github[81].

## Transmission electron microscopy

For transmission electron microscopy, *L. mexicana* were fixed directly in medium for 10 min at room temperature in 2.5% glutaraldehyde (glutaraldehyde 25% stock solution, EM grade, Electron Microscopy Sciences). Centrifugation was carried out at room temperature for 5 min at 16,000 × g. The supernatant was discarded and the pellet was fixed in 2.5% glutaraldehyde and 4% PFA (16% stock solution, EM grade, Electron Microscopy Sciences in 0.1 M PIPES (pH 7.2)) for minimum 2 h. Cells were embedded in 3% agarose and contrasted with $OsO_4$ (1%) (osmium tetroxide 4% aqueous solution, Taab Laboratories Equipment) for 2 h at 4 °C. Cells were stained with 2% of Uranyl Acetate for 2 h 4 °C. After serial dehydration with ethanol solutions, samples were embedded in low-viscosity resin Agar 100 (Agar Scientific, UK) and left to polymerise at 60 °C for 24 h. Ultrathin sections (90 nm) were collected on nickel grids using a Leica EM UC7 ultramicrotome and stained with uranyl acetate (1%, w/v) (uranyl acetate dihydrate, Electron Microscopy Sciences) and Reynolds lead citrate[82] (80 mM Lead nitrate (Thermofisher, L/1450), 120 mM sodium citrate (Sigma, 71405) and 1 M sodium hydroxide (SLS, CHE3422). Observations were made on a Thermo Fisher Scientific Tecnai12 or JEOL 2100 Plus 200 kV transmission electron microscope with a Gatan OneView camera.

## Ninefold rotational averaging of *L. mexicana* axonemes

For generation of averaged axoneme views, doublet A tubule centres in axoneme images were identified, fitted to an ellipse then perspective corrected to ensure circularity, followed by ninefold rotational averaging[83]. Twenty-five rotationally averaged axonemes were then aligned and averaged. Difference maps were generated by comparison with the average 25 rotationally averaged parental cell line, and per-

pixel statistical significance of electron density changes calculated by Mann Whitney U test (with multiple-comparison correction for the number of pixels within the axoneme cross-section).

**Protein structure prediction and comparison**
Protein structures were predicted using AlphaFold2[53] for the *T. brucei* ortholog with lineage-optimised input multiple sequence alignments[54,55,84]. Predicted secondary structure (α helix and β sheet) were taken from the cif file output. For proximal and distal paralogous pairs, sequence similarity was measured from a Clustal Omega[85] sequence alignment of the *T. brucei* protein sequences, counting identical positions divided by longest protein sequence length. Structure similarity was measured by FATCAT flexible alignment[86,87] of the predicted structures, reporting FATCAT similarity *p* value and root mean squared deviation (RMSD) of equivalent aligned positions. For protein phylogenetics, members of the protein family were identified by BLASTp from the *T. brucei* protein against the reference kineto-plastid genomes in TriTrypDB[72], with $e < 10^{-3}$, and phylogeny inferred using ETE3 with default settings[88].

**Statistics and reproducibility**
All microscope images shown are representative of one genetically validated cell line, selected from images of several hundred example cell images unless otherwise indicated in the figure legend. Sample types, sample sizes and statistical tests are described in figure legends.

**Reporting summary**
Further information on research design is available in the Nature Portfolio Reporting Summary linked to this article.

## Data availability
Data supporting this study are presented within the paper and its supporting information. Source data are provided with this paper.

## Code availability
Custom code used in this study are made available as described in materials and methods (https://github.com/Mar5bar/flagella-analysis-pipeline).

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

## Acknowledgements

We thank Dr. Errin Johnson and Dr. Charlotte Melia for technical assistance for transmission electron microscopy (Electron microscopy facility at Sir William Dunn School of Pathology, Oxford University, United Kingdom). This work was supported by a Wellcome Trust Sir Henry Dale Fellowship [211075/Z/18/Z] awarded to R.J.W. B.J.W. is supported by the Royal Commission for the Exhibition of 1851.

## Author contributions

R.J.W. conceived and supervised the project. L.B. and C.F. measured fluorescence signal distribution along the length of flagella from epi-fluorescence micrographs. C.F. made all the cell line and performed PCR validation. C.F. performed motility assays, recorded flagellum beat movies, electron microscopy and analysed experiments. R.J.W. performed electron microscopy image analysis. B.J.W. and R.J.W. wrote flagellar beating analysis scripts. C.F. and R.J.W. wrote the first draft of the paper and all authors reviewed, edited and commented on the draft.

## Competing interests

The authors declare no competing interests.
