## [Transparent Peer Review file · Nature Communications]

Proteins with proximal-distal asymmetries in axoneme localisation control flagellum beat frequency

Corresponding Author: Dr Richard Wheeler

Version 0:

Reviewer comments:

Reviewer #1

(Remarks to the Author)

Proteins with proximal-distal asymmetries in axoneme localisation control flagellum beat frequency

Fort et al.

The work described in the manuscript provides a comprehensive analysis of the proximal-distal asymmetry of axonemal proteins in *Trypanosoma brucei* and *Leishmania mexicana*, identifying several proteins with conserved asymmetric localization and investigating their functional roles in axonemal structure and flagellar beat regulation. The study employs various techniques, including endogenous protein tagging, deletion mutants, electron microscopy, and beat waveform analysis, to elucidate the molecular mechanisms underlying axoneme organization and function. Overall, it is a very comprehensive, well-controlled, and impressive study of the newly discovered asymmetries in axonemal distribution of flagellar proteins.

I have a few main concerns regarding the conclusions of the paper and suggestions for further approval before it can be accepted in Nature Communications:

Major comments

1. The authors cleverly exploit the unique availability of globally fluorescent tagged proteins in the TrypTag database to survey for proteins enriched in *T. brucei* cilia. As a control the identified cilia proteins are assessed and validated for in the *L. mexicana* proteome. To identify conserved motile cilia protein functions between *T. brucei* and *L. Mexicana*, the protein sequence analysis was performed using BLAST. While BLAST has for many years been the golden standard for sequence database searches, today there are much better and sensitive search algorithms for phylogenetic analysis. Those search algorithms are based on hidden Markov model (HMM) searches often reveal many protein relationships not identified by BLAST. The authors identify 25 *L. mexicana* orthologs of 55 *T. brucei* cilia proteins. This seems like a limited number of orthologs. The authors would likely identify several additional homologies if more sensitive search approaches were used in their phylogenetic analysis. It would be interesting to see which orthologs/paralogs are identified between *T. brucei* and *L. Mexicana* using for instance HHblits (<https://toolkit.tuebingen.mpg.de/tools/hhblits>) or Jackhmmer (<https://www.ebi.ac.uk/Tools/hmmer/search/jackhmmer>). Importantly, such remote homology searches could also reveal distant homologies to hitherto unknown human cilia proteins, which would appeal to a broader audience.

2. In Figure 1 the quantification of fluorescent protein cilia localization is shown. It is a great to see the careful examination and correlation of protein cilia localizations using both N- and C-terminally tagged proteins. However, some important information is missing. For instance, how was this quantification performed? In addition, how were the "4 protein localization groups" categorized? Some statistical analysis e.g., cluster analysis could perhaps be used.

3. The "short proximal" localization proteins are discussed concerning Figure 1B, F. What does this localization represent in the cilia architecture? How is it measured differently from e.g., the cilia transition zone? Some cilia markers could be used to co-stain the individual cilia subcompartments. Co-localization with proteins that defines such cilia subcompartments would

support the authors conclusions regarding the “short proximal location” group and “4 protein localization groups”. ARL13B is well known to be distributed across the entire cilium in e.g., human cilia with exceptions as mentioned in the introduction. It is therefore also interesting that the authors find that *T. brucei* ARL13B shows confinement to their claimed “short proximal” region. Is this an artifact of the model organism used or is this observation genuine? Other well-known pan-cilia markers should be tested to stain *T. brucei* and *L. mexicana* cilia. How would that manifest in the fluorescent quantification of cilia? In the results presented in Figure 3A, it is shown that deletion of pDC1 does not affect the “short proximal” localization proteins, as also noted the authors. Hence, while the authors favor a conclusion that there are two mechanisms for cilia proximal protein localization, this result could instead be explained by limitations in the fluorescent protein detection.

4. The authors further examine whether DC-dependent asymmetrically positioned proteins are ODA-associated using the above-mentioned physiological assays. While the fluorescent microscopy assays used in this study are elegant, when it comes to assessing specific protein-protein associations, microscopy (fluorescence or electron) would not normally be the first and best choice. At best microscopy can support biochemical characterization of protein complexes. Hence to better strengthen the conclusions of this study that e.g., FLAM6, LmxM.36.5300, ARL13B, LC4-like, and LmxM.30.0090 bind to an ODA heavy chain, affinity purification assays using these proteins as bait followed by mass spectrometry analysis should be tested. Could be FLAG/Neogreen IP's. To test these interactions specifically inside the cilia compartment, BioID-tagging (e.g., ultraID/microID) and proximity labeling of one or more endogenous cilia proteins could be tested. The protein size of ultraID is ca. 20 kDa which is comparable to the size of neogreen. Such experiments could also provide more detailed insight into the observed different mechanisms involved in proximal-distal asymmetry of ODA-associated proteins.

5. Figure 8. The biochemical properties presented in Figure 8 do not indicate homologies between orthologs or provide the interesting information. Namely, the phylogenetic sequence conservation and tertiary structure. It would be more informative to show multiple sequence alignments with statistical score values. If AlphaFold2 is used then protein 3D fold superimpositions between orthologs could be shown.

Minor comments

1. Revise the “Protein sequence analysis” method section. It does not read well and there are many errors.
2. Many important details for reproducibility are missing in the “Genetic modification” method section. The sequences of sgRNA should be provided and a brief description of the design and PCR protocol should be added instead of just referring to other work.
3. In the “Flagellar beating analysis” method section the computation protocol description is not clear “A range of beating characteristics were computed for each cell and all deletion cell lines were compared to the parental cell line. Full code for the analysis pipeline (including thresholds for exclusion) is available on request”. The script should be made available with a link (a requirement of this journal) so that readers can reproduce the analysis.
4. Technical Details and Reproducibility. Ensure that all experimental procedures, especially those related to protein tagging and deletion generation, are described in sufficient detail for reproducibility. This includes specific oligo sequences used for tagging, details of PCR conditions, and any controls employed.
5. Typo in the introduction. “The outer dynein arms (ODAs) attached to the nine doublets every 24 nm and are canonically...”. The outer dynein arms (ODAs) ARE attached...
6. Typo in the introduction. “Overall, this suggests interplay of signalling and asymmetric protein distribution”. Replace with “Overall, this suggests AN interplay of signalling ...”
7. Typo in the Results section. “However, the localisation of PDEB2::mNG, a representative distal enriched PDEs, was unaffected (Figure 3D).” Change to “a representative distal enriched PDE, was ...”

Reviewer #2

(Remarks to the Author)

Cecile Fort et al. present an insightful study on the proximal-distal asymmetry of the flagellum in *Leishmania mexicana*, a well-characterized flagellated parasite and a model system for examining flagellar assembly and function. Building on previous findings of proximal-distal asymmetry in the Outer Dynein Arm-docking complex (ODA-DC) within the flagellum of *Trypanosoma brucei* (Edwards et al. PNAS 2018), the authors utilized the *T. brucei* genome-wide protein tagging database TrypTag to identify 55 proteins with proximal-distal asymmetric localization.

Recognizing the importance of conserved components across species, these proteins were cross-referenced with orthologs in *L. mexicana*, retaining 15 proteins exhibiting comparable localization asymmetries in both species. These selected proteins underwent detailed analysis in *L. mexicana* to elucidate their roles in flagellum structure and function. By measuring the fluorescence signal distribution along the axoneme, distinct proximal-distal distribution patterns were revealed. Proteins with similar localization patterns were classified as proximal DC-like (pDC-like) or distal DC-like (dDC-like), and additional patterns such as “short proximal” and “distal enriched PDEs” were identified.

Through the generation of deletion mutants, the authors evaluated the role of these proteins in flagellar assembly and

function, assessing flagellar length, ultrastructure via electron microscopy (EM) averaging densities, swimming behavior, and flagellar beat patterns. The study explored the dependency of proximal and distal-specific protein localization on DC asymmetry. It was found that proteins with pDC-like localization depended on the presence of pDC1 and pDC2, while short proximal proteins were unaffected by pDC1 deletion. Similarly, the deletion of dDC2 impacted the localization of dDC-like proteins but not distal enriched PDEs, indicating the presence of two distinct mechanisms for proximal and distal-specific localization in trypanosomatids. Interestingly, deletions of certain proximal-specific proteins affected the overall beat frequency without altering the structural integrity of the flagellum, suggesting a regulatory role in flagellar beat modulation.

Flagella and cilia are critical for the motility and sensory functions of many eukaryotic cells, including the pathogenic trypanosomatids *Trypanosoma brucei* and *Leishmania mexicana*. The structural and functional asymmetry of flagella is vital for generating effective beating patterns necessary for locomotion and host infection. Thus, understanding the molecular architecture of the flagellum is essential for elucidating the mechanisms underlying flagellar beating.

This comprehensive analysis of proximal-distal asymmetry in flagellar proteins between *T. brucei* and *L. mexicana* underscores the conservation and functional significance of these asymmetries. The study identifies distinct localization mechanisms and provides insights into the regulation of flagellar beating. These findings advance our understanding of the conservation and functional implications of these asymmetries across different species, enhancing our knowledge of flagellar biology.

Overall, this is an excellent article. The text is well-written, and the data are presented clearly and logically. The findings appear robust and support the conclusions. This work is highly significant for the broader community studying flagella and cilia structure and function, offering numerous hypotheses to be tested in other systems, particularly in patients with ciliopathies.

Major Comments:

The asymmetry in proximal-distal localization of proteins along flagellar axonemes has been previously described, notably in *Trypanosoma* by the authors, as well as in *Chlamydomonas* and humans. In *Chlamydomonas*, more extensive structural heterogeneities among microtubule doublets (MTDs) were observed in the proximal region of the axoneme, with one microtubule doublet exhibiting particularly strong proximal/distal asymmetry. The study presented here is already highly innovative, and incorporating recent advancements in super resolution microscopy could further enhance its novelty and impact, making it an even stronger candidate for high-impact journals such as *Nature Communications*. Expansion microscopy significantly enhances resolution and allows for high-resolution mapping of proteins relative to microtubule doublets. It could provide a more detailed understanding of protein localization within the axoneme, potentially uncovering new insights into the mechanisms of proximal-distal asymmetry. For example, recent studies by Laporte et al. *Cell*. 2024 have demonstrated the effectiveness of expansion microscopy in revealing intricate details of centriole assembly.

While this study provides a systematic analysis of proteins with asymmetric localization in axonemes, it would benefit from a discussion on the role of this asymmetry across different stages of the parasitic life cycle. Specifically, *Leishmania* transitions into an intracellular amastigote form, which features a very short but motile flagellum, in contrast to *Trypanosoma brucei*, which lacks an intracellular stage. A brief discussion on this aspect would be particularly valuable for understanding the *L. mexicana* orthologs that do not exhibit asymmetry in their localization in the study (what about orthologs in *T. cruzi*, which also adopts an amastigote form with a short motile flagellum?). This could provide critical insights into the functional and evolutionary adaptations of flagellar proteins in relation to the distinct life cycle stages and cellular environments of *Leishmania* compared to *Trypanosoma brucei*.

Minor comments:

page 2: Correct the spelling in "including a new paralogous."

page 3:

- Provide a table listing the 55 identified *T. brucei* proteins for easier reference.
- Consider presenting the first three columns of Table S4 earlier in the manuscript and consistently use either the protein names or accession numbers throughout the text and figures. It would be much easier for the reader to track the various proteins.
- Clarify "leaving 10 proteins for detailed analysis." The text suggests 15 proteins were retained, with 14 further characterized.
- In the Table S2 legend, specify what "enriched proteins" means (compared to what?).
- For Figure S1, indicate whether the proteins were also N-terminal tagged to control for asymmetric localization.
- In Figure S4, confirm the deletion of LmxM.30.0090 because a higher band is observed and the PCR control (PF16) did not work.

page 4:

- "As these included Δ pDC2, we believe these noisy observations are spurious, perhaps due to small variability in doublet position (Figure 2D, 2E)."
These observations could be influenced by factors not yet identified, and exploring these potential impacts may provide further insights."

page 5:

- PDEB2 localization was unaffected in the DC2 deletion mutant. The thorough analysis of PDEB2 localization is commendable. Additionally, assessing whether PDEB1 is affected could further elucidate the differential roles of PDEB1

and PDEB2, adding another layer of depth to this already insightful study.

- In Figure 4A, consider reordering the images to present the $\Delta ODA\alpha$ mutant first, as it aligns better with the text.

page 7:

- Remove "that did" in "however, those that did had a significantly increased beat frequency."

- For Figure 7B,E, note that $\Delta LC4$ -like, $\Delta PDEB2$, and $\Delta FLAM6$ display a higher proportion of cells with base-to-tip beating. Do the authors have a hypothesis for this observation?

- For Figure 7C, clarify if the round dots represent the average. If so, $\Delta PDEB2$ does not display a significant increase in beat frequency but more variability, suggesting less control.

page 8

- Revise "We previously noted that ODA proximal-distal asymmetries occur across diverse eukaryotes." It would be insightful to discuss whether the characterized proteins in this study are conserved across other organisms and provide their IDs. This could enhance understanding of proximal-distal asymmetry in axonemes.

page 9:

- "The PFR-free and FAZ structures broadly line up with the short-proximal region, but dependence of PFR or FAZ

positioning on axoneme asymmetries is unlikely as we saw no obvious change to the PFR or FAZ by electron microscopy in the SPA1 and SPA2 deletion mutants."

The converse was not tested, so the conclusion should be toned down.

page 10:

- Correct "We can consider a two key hypotheses" to "We can consider two key hypotheses."

- Correct the spelling of "increasesing"

- Remove "Domain identification, ortholog identification, structure prediction, etc. Cite tritrypdb, cite my alphafold."

page 11:

- Correct "We used a positive control of with genomic DNA from the parental cell line" to "We used a positive control with genomic DNA from the parental cell line."

- Reformat "(Wheeler, 2017)."

- Correct the spelling of "1uL of 5 μ m polystyrene beads."

- Specify "Cells with only one observable flagellum" rather than "Cells with one flagellum (non-dividing)," as it is difficult to determine if a tiny new flagellum has not started to grow early in the cell cycle.

page 12:

- Specify the concentrations used for "Reynolds lead citrate71 (Lead nitrate (Thermofisher, L/1450), sodium citrate (Sigma, 71405) and sodium hydroxide (SLS, CHE3422)".

page 31

- Clarify the meaning of the asterisks in Table S4

Reviewer #3

(Remarks to the Author)

In this strong and important paper, the authors complete a systematic identification of proteins with proximal-distal asymmetry in the axonemes of Trypanosoma and Leishmania species, indicating this phenomenon applies to more proteins than previously anticipated. Functional analysis and assessment of interdependencies among asymmetrically localized proteins identified at least two different mechanisms for achieving asymmetry, while examining a multitude of flagellum beat parameters demonstrated at least some of the proteins identified are required for normal flagellar beating.

The work is rigorous and the paper well-written. It represents an important contribution because, while longitudinal symmetries have been reported in several organisms, the extent of this phenomenon and underlying mechanisms remain poorly understood. Moreover, the requirement for such asymmetry in flagellar beating is poorly defined. This work therefore provides advances on multiple levels and provides a foundation for further dissecting the impact of and mechanisms for establishing longitudinal asymmetries in the axoneme. I have a few suggestions below for the authors that I think will improve the paper:

1. The DISCUSSION is a bit long and could be shortened, as this will improve the impact and reach of the paper, particularly for readers outside the direct area of flagellum biology.

2. On p5 of 43, the sentence "...mutation of ...ODA4..." should be corrected as less general. Note that some mutations at the oda4 (beta HC) alter ODAalpha, but others do not. e.g J Cell Biol. 1993 Aug 1; 122(3): 653-661.

3. p7 of 43, last line of RESULTS.

-I advise removing the term "regulators", as this implies the protein may switch beat parameters in response to a signal. Results indicate loss of some proteins influences beat parameters, but a role as a "regulator" has not been assessed.

4. Fig 7.

--please define "dominant" frequency.

--The multi-factorial analysis of motility and flagellar beating is to be commended. Note, however, that it is hard to reconcile some of the swimming speed results with results of waveform analysis. For example, delta-pDC2 shows a dramatic increase in "uncoordinated" beating (now >60% of cells compared to ~5% in WT), yet no substantial change in swim speed. It is hard to imagine uncoordinated beating driving normal swimming speed. I suspect there is a fair amount of noise in the waveform analysis, so perhaps one can conclude general impact on flagellum beating, but care should be taken in drawing more specific conclusions. Please clarify/comment regarding this.

5. I struggle a bit to distinguish "short proximal axoneme" from "pDC-like", as it seems both are defined as being in the proximal 20% of the axoneme. Please clarify.

6. There is a recent paper examining mechanisms of proximal/distal asymmetry in trypanosoma that needs to be cited and discussed: Bonnefoy, S., Alves, A. A., Bertiaux, E. & Bastin, P. LRRC56 is an IFT cargo required for assembly of the distal dynein docking complex in *Trypanosoma brucei*. *Mol. Biol. cell* 35, ar106 (2024).

7. Fig 8.  in panel A, dDC is mislabeled as pDC

Version 1:

Reviewer comments:

Reviewer #1

(Remarks to the Author)

Comments to the revised manuscript: Proteins with proximal-distal asymmetries in axoneme localisation control 1 flagellum beat frequency

The authors have made significant revisions based on the reviewers' feedback. To summarize.

They revised multiple figures, added a new supplemental figure, re-analyzed data, and incorporated double-deletion mutants. They expanded the methods section, introduced hierarchical clustering to strengthen conclusions on protein localization, and improved reporting of experimental details (e.g., oligonucleotide sequences and analysis pipeline code). They employed enhanced ortholog detection using Jackhmmer but retained their original OrthoMCL/BLAST approach for sensitivity. In addition, They performed new quantitative analyses to differentiate protein localization in axonemal structures from transition zones.

Finally, they streamlined the discussion, clarified key points, and consolidated supplemental tables for improved clarity.

The authors have performed a thorough revision of their initial manuscript and added new data to support their conclusions and in some instances made corrections to the initial conclusions. I have the following major comments to the revised manuscript.

1. The authors reassessed the protein sequence homologies between *T. brucei* and *L. Mexicana* using Jackhmmer. In the rebuttal they state that Jackhmmer fails to detect several syntenic orthology pairs and decides to their existing method using a combined OrthoMCL/BLAST approach. To support their conclusion, they provide a list of syntenic pairs that Jackhmmer fails to detect. I have a problem with that analysis as I can see that several of the proteins provided in the list bear highly conserved and redundant protein domains (e.g., in the case of Tb927.11.16840 and other). In such cases, sensitive algorithms for phylogenetic analysis e.g., Jackhmmer or HHblits are not suitable as they produce much too many matches which makes phylogenetic analysis impossible. In that case, single BLAST searches based on overall sequence percentage homology are a better choice. Hence using Jackhmmer in this case does not prove any point. The HMM-based search tools are useful in all the other cases where only a few matches are found using BLAST. In those cases, usually, HHblits is the method of choice to expand the MSA and then convert that to an HMM that can be used to search against e.g., the proteome of an organism of choice. I suspect that if that sensitive approach was used (specifically in those borderline cases) more matches would be found. This could be considered for future phylogenetic analysis.

2. In the Figure 1E the authors present the new Hierarchical clustering of signal intensity profiles for flagellar asymmetrically distributed proteins. It was a bold claim in the initial version of the manuscript that four protein localization groups were identified considering that the groups were manually assigned. It is great to see that the authors employed hierarchical clustering of signal intensity profiles and found several new what I guess are "localization groups" (now 14 proximal and distal clusters identified) but now termed classes. But now that multiple groups/classes have been identified this begs the question. What is the physiological significance of all these clusters (groups)? Is it at all likely that so many compartments exist inside the flagellum? What if that twice the number of clusters (28) had been identified instead? Could the several clusters identified reflect the possibility that these intraflagellar proteins are spatially and temporarily dynamic rather than occupying specific localizations in flagellar? That is, do these many clusters reveal a weakness in the measuring approach? There should be some reflections about these concerns in the discussion. The authors note that many axoneme-localizing proteins localize along the entire flagellum as expected. However, these proteins are known not to be dynamic in terms of localization. It is another problem dealing with dynamic proteins that might switch places continuously.

3. I asked the authors to provide more biochemical evidence for their proposed intra cilia protein associations e.g., using

affinity purification followed by proteomics analysis. In their response, the authors ask hypothetically “How would evidence for the interaction of LC4-like with an ODA heavy chain explain why this interaction only occurs in the distal flagellum? How would interaction of a novel protein with the proximal docking complex show that the novel protein is proximal specific, and preclude similar interaction with the distal docking complex?”.

This biochemical evidence is important because most likely different cilia subcompartments reflect different subcomplexes as has been established for other cilia compartments e.g., the centrioles, basal body, transition zone, and the central pair. These scientific terminologies are justified because their existence is recapitulated by completely different methodologies i.e., imaging and biochemistry.

4. In their rebuttal, The authors state, "Because proteomic analysis of a proximity labeling, or affinity purification is not spatially resolved* it does not answer our primary question."

This is not true. Proximity labelling proteomics is in fact routinely used today in cell biology research exactly because it can detect spatially distinct protein niches/compartments in the cell. A publication in Nature Communications, I believe, requires evidence at the protein level to support any claim of a new “protein association” as suggested in the first version of the manuscript. This is more imperative now that the initial four “localization groups” have been ruled after more careful examination in the revised manuscript.

(Remarks on code availability)

Reviewer #2

(Remarks to the Author)

Dear Editor,

Thank you for the opportunity to review the revised manuscript from C. Fort et al. entitled “Proteins with proximal-distal asymmetries in axoneme localisation control flagellum beat frequency”.

I appreciate the authors' effort in addressing my comments. Overall, they have successfully tackled most of the issues I raised, and I believe the manuscript has improved as a result.

The authors have successfully clarified several points that were previously ambiguous and have provided additional data that strengthens their conclusions.

The manuscript has been restructured in parts to improve the flow and readability, which is beneficial.

Note : In Figure 2E and Figure 8: the panels are cut off and not fully visible. Table S3 is not mentioned in the text.

Aside from these minor issues, I am satisfied with the authors' revisions and recommend acceptance of the manuscript.
Best regards

(Remarks on code availability)

Reviewer #3

(Remarks to the Author)

The authors have provided a very thoughtful and thorough revision that addresses reviewer concerns. The paper is improved and a nice piece of work

(Remarks on code availability)

Version 2:

Reviewer comments:

Reviewer #1

(Remarks to the Author)

The authors have provided a detailed and thoughtful response to my previous comments. Their revisions effectively address the concerns raised, and the manuscript has improved as a result. The clarifications and adjustments made strengthen the overall quality of the work, making it a solid contribution to the field

(Remarks on code availability)

Reviewer #1

The work described in the manuscript provides a comprehensive analysis of the proximal-distal asymmetry of axonemal proteins in *Trypanosoma brucei* and *Leishmania mexicana*, identifying several proteins with conserved asymmetric localization and investigating their functional roles in axonemal structure and flagellar beat regulation. The study employs various techniques, including endogenous protein tagging, deletion mutants, electron microscopy, and beat waveform analysis, to elucidate the molecular mechanisms underlying axoneme organization and function. Overall, it is a very comprehensive, well-controlled, and impressive study of the newly discovered asymmetries in axonemal distribution of flagellar proteins.

I have a few main concerns regarding the conclusions of the paper and suggestions for further approval before it can be accepted in Nature Communications:

Major comments

1. The authors cleverly exploit the unique availability of globally fluorescent tagged proteins in the TrypTag database to survey for proteins enriched in *T. brucei* cilia. As a control the identified cilia proteins are assessed and validated for in the *L. mexicana* proteome. To identify conserved motile cilia protein functions between *T. brucei* and *L. Mexicana*, the protein sequence analysis was performed using BLAST. While BLAST has for many years been the golden standard for sequence database searches, today there are much better and sensitive search algorithms for phylogenetic analysis. Those search algorithms are based on hidden Markov model (HMM) searches often reveal many protein relationships not identified by BLAST. The authors identify 25 *L. mexicana* orthologs of 55 *T. brucei* cilia proteins. This seems like a limited number of orthologs. The authors would likely identify several additional homologies if more sensitive search approaches were used in their phylogenetic analysis. It would be interesting to see which orthologs/paralogs are identified between *T. brucei* and *L. Mexicana* using for instance HHblits (<https://toolkit.tuebingen.mpg.de/tools/hhblits>) or Jackhmmer (<https://www.ebi.ac.uk/Tools/hmmer/search/jackhmmer>). Importantly, such remote homology searches could also reveal distant homologies to hitherto unknown human cilia proteins, which would appeal to a broader audience.

Firstly, to clarify, to identify *L. mexicana* orthologs of *T. brucei* proteins we used a combination of syntenic orthologs supplemented with reciprocal best BLASTP searches. The former were derived from the genome database TriTrypDB (which uses OrthoMCL) and we apologise for this not being clear in the text (see also minor comment 1). We have updated the methods text to clarify this, and have updated the supplemental table (now consolidated) of genes to indicate how orthology was discovered.

We agree that the number of clear single orthologs of *T. brucei* proteins is perhaps surprising, and as a key result of interest we agree that more analysis here is well justified. Similarly, enhanced detection of potential human orthologs is always beneficial.

As suggested, we used Jackhmmer to carry out the reciprocal search between *T. brucei* and *L. mexicana*. Specifically, to maximise sensitivity, for each gene we used phmmer to search against UniRef50, built a hidden Markov model from the life-wide search result, then searched using this model against the *L. mexicana* genome. Then we did the reciprocal search using the same method of the top *L. mexicana* hit against the *T. brucei* genome, and checked if a reciprocal match. This approach fails to detect several syntenic orthology pairs:

Tb927.10.5230	LmxM.36.0820
Tb927.11.16840	LmxM.31.3200
Tb927.7.4600	LmxM.14.0100
Tb927.10.6570	LmxM.36.2070
Tb927.3.4270	LmxM.08_29.1700

Therefore, we have retained the existing method as the combined OrthoMCL/BLAST approach appears more sensitive.

We have revisited the search for human and *Chlamydomonas* orthologs. We had previously searched using reciprocal best BLAST, and enhanced this using reciprocal best Jackhmmer (although that gave no more hits). This mapping of kinetoplastid biology to humans and *Chlamydomonas* has been added to Table S1, and the key result that orthologs of pDC4/dDC4 and SPA2 are present in *C. reinhardtii* has been clarified in the discussion.

2. In Figure 1 the quantification of fluorescent protein cilia localization is shown. It is a great to see the careful examination and correlation of protein cilia localizations using both N- and C-terminally tagged proteins. However, some important information is missing. For instance, how was this quantification performed? In addition, how were the “4 protein localization groups” categorized? Some statistical analysis e.g., cluster analysis could perhaps be used.

We apologise for the absence of detail – there was no advanced methodology used here, simply a manual trace of the flagella and measurement of pixel intensity values, then Excel wrangling to normalise and plot the data. We have added an expanded methods section to explain this. (The reviewer lists the methodological detail as a ‘for instance’, what else was missing?)

The 4 protein localisation groups were manually assigned, however as suggested, to add robustness, we have added a clustering analysis. *Ab initio* identification of number of groups/clusters in a dataset lacks an unambiguously correct computational solution, therefore we carried out a hierarchical clustering analysis. This cannot define the number of groups we should use, but does map similarity. This showed that N and C terminal tagging typically give quantitatively similar localisations (detailed in the updated text), and that our manual classification is broadly quantitatively supported. It also led us to remove one protein, Tb927.11.3920, as too similar to a whole axoneme signal to justify its inclusion. This analysis has been added to Figure 1.

In combination with the point below, we can see some concern about drawing biological conclusions from only localisation data so have carefully described the localisations within each cluster rather than using biologically-based (eg. pDC-like, PDEB-like) names. We now refer to the clusters primarily in this more abstract way, and have largely removed group labels from figures to avoid over-interpretation of these data. The full set of classifications are, however, listed in Figure 1 and Table S1.

3. The “short proximal” localization proteins are discussed concerning Figure 1B, F. What does this localization represent in the cilia architecture? How is it measured differently from e.g., the cilia transition zone? Some cilia markers could be used to co-stain the individual cilia subcompartments.

Ensuring that these localisations are distinct from known proximal compartments, ie. the transition zone, is an important point. We had made the decision based on visual appearance as an extended line: we and others have shown that proteins which localise to

the transition zone are typically confined to a small point (eg. doi:10.1016/j.molbiopara.2018.12.003).

However, we can see the value of a quantitative test for transition zone proteins. To do so, we carried out measurement of the start of the green signal from the centroid of the kinetoplast (mitochondrial DNA) in *T. brucei*. The kinetoplast is held at a fixed distance from the basal body by the tripartite attachment complex, making this a robust measure, shown by the accurate measurement of the position of known basal body and transition zone markers. This showed that there were no transition zone proteins: Tb927.10.4720 and Tb927.11.5770 signals started closest to the kinetoplast, but were still similar to the position of the basal plate (basalin marker, doi:10.7554/eLife.42282) thus are bona fide axoneme components. This analysis did, however, identify Tb927.9.2075 as starting notably far from the kinetoplast, consistent with a flagellum attachment zone protein. Indeed, it has now been characterised as one – this has gene therefore been removed from the corresponding supplemental figure.

This additional quantitative analysis has been added to Figure 1.

Co-localization with proteins that defines such cilia subcompartments would support the authors conclusions regarding the “short proximal location” group and “4 protein localization groups”.

Co-localisation would provide some additional information, but co-localisation of proteins does not provide evidence for association with the same underlying structure/mechanism which generates a particular localisation group – which was our ultimate aim. Indeed, the new hierarchical clustering points to more than 4 localisation groups. Evidence for a localisation group is better summarised by raw data, ie. the new clustering analysis, or completely orthogonal functional data – all conclusions we draw are already and better supported by the tests for physiological dependence for assembly in Figures 3, 5 and 6.

To address this point, we have updated the text and figures to avoid declarative statements about number of groups, and instead let the data speak for itself – especially the dependency for assembly.

ARL13B is well known to be distributed across the entire cilium in e.g., human cilia with exceptions as mentioned in the introduction. It is therefore also interesting that the authors find that *T. brucei* ARL13B shows confinement to their claimed “short proximal” region. Is this an artifact of the model organism used or is this observation genuine?

The proximal flagellum-specific localisation of ARL13B in *T. brucei* is not a new result, it was previously shown 6 years ago using endogenous tagging and a native antibody (doi:10.1242/jcs.219071). We've extended this using N and C terminal tagging and tagging in a related (but nonetheless separated by ~100 million years) species. Given this result is observed by two research groups in two trypanosomatid species by two independent methods it certainly appears genuine.

I'm afraid it is unclear what is meant by “an artifact of the model organism used”? Using the same phraseology, could the full-length flagellum localisation in humans be ‘an artifact of higher primates’ or perhaps ‘an artifact of short primary cilia’?

We have added some additional detail to the introduction at the first mention of ARL13B to clarify that it does normally localise to the full ciliary length in humans, but also the specific evidence for asymmetric distribution in mice and *T. brucei*.

Other well-known pan-cilia markers should be tested to stain *T. brucei* and *L. mexicana* cilia. How would that manifest in the fluorescent quantification of cilia? In the results presented in Figure 3A, it is shown that deletion of pDC1 does not affect the “short proximal” localization proteins, as also noted the authors. Hence, while the authors favor a conclusion that there are two mechanisms for cilia proximal protein localization, this result could instead be explained by limitations in the fluorescent protein detection.

Please bear in mind that this analysis is based on data from our genome wide protein localisation dataset (doi:10.1038/s41564-022-01295-6) – if any well-known proteins had an unexpected proximal or distal-specific signal then they would have been identified and listed in Figure 1. By their absence, this means that all radial spoke, nexin link, central pair, inner dynein arm, outer dynein arm etc. do indeed localise to the full length of the flagellum. The docking complex is prominent as the only exception. We have more clearly stated this in the text, to convey the robustness of our survey. This also suggests that protein tagging very rarely causes spurious asymmetric localisations.

To address your specific concern about the quantitative analysis of a “pan-cilia” marker fluorescent signal, we have added quantitation of two radial spoke proteins: RSP9 and RSP9-like. We recently characterized this as a paralogous pair which, unlike p/dDC1/2/4, does not have proximal-distal asymmetry in localisation (DOI:10.1242/jcs.260655) making it a particularly powerful example. These data can be seen in Figure 1.

We do recognise that the localisation determined by endogenous tagging may be altered by the presence of the tag, and we have added a clear statement to the text that localisations observed are that of the experimental fusion protein and may not completely match the untagged protein. This is done next to the “two mechanisms” conclusion in the results, to make this caveat as clear as possible.

4. The authors further examine whether DC-dependent asymmetrically positioned proteins are ODA-associated using the above-mentioned physiological assays. While the fluorescent microscopy assays used in this study are elegant, when it comes to assessing specific protein-protein associations, microscopy (fluorescence or electron) would not normally be the first and best choice. At best microscopy can support biochemical characterization of protein complexes. Hence to better strengthen the conclusions of this study that e.g., FLAM6, LmxM.36.5300, ARL13B, LC4-like, and LmxM.30.0090 bind to an ODA heavy chain, affinity purification assays using these proteins as bait followed by mass spectrometry analysis should be tested. Could be FLAG/Neogreen IP's. To test these interactions specifically inside the cilia compartment, BioID-tagging (e.g., ultraID/microID) and proximity labeling of one or more endogenous cilia proteins could be tested. The protein size of ultraID is ca. 20 kDa which is comparable to the size of neogreen. Such experiments could also provide more detailed insight into the observed different mechanisms involved in proximal-distal asymmetry of ODA-associated proteins.

Such techniques were considered as a discovery tool over the course of the project, however we did not use them as they could not readily answer our questions about generation of asymmetry. Perhaps we are misunderstanding your comment, but consider

the following hypothetical examples: How would evidence for interaction of LC4-like with an ODA heavy chain explain why this interaction only occurs in the distal flagellum? How would interaction of a novel protein with the proximal docking complex show that the novel protein is proximal specific, and preclude similar interaction with the distal docking complex? Because proteomic analysis of a proximity labelling or affinity purification is not spatially resolved* it does not answer our primary question.

We agree that affinity purification and proteomic methods provide a gold standard for identifying strong protein-protein interactions, but we did not aim to make claims about specific biochemical complexes. Instead, we focused on the biological consequence of any interaction, ie. where proteins ultimately end up in the flagellum, being careful to describe this in terms of assembly dependencies, and have carefully checked that the text is clear in this regard.

We agree affinity purification experiments could “provide more detailed insight into the observed different mechanisms” but, as the reviewer indicates, these are detailed biochemical studies which would better fit in future work about specific proteins.

* It is worth noting that we did try to design complex sample preparation approaches to fractionate the flagellum and give this type of information, however we were unable to design a compelling experiment. Specifically, we wanted to avoid harsh treatments to extract flagellar complexes from the axoneme (which risk disrupting more weakly-bound components). We also needed a method to circumvent the observation that our previous proteomics of purified *Leishmania* flagella (doi:10.1371/journal.ppat.1007828) failed to detect the proximal docking complex (suggesting that purified flagella are sheared some distance from their base). Finally, we also needed to exclude any preassembled cytoplasmic complexes (particularly expected for outer dynein arms, and which may be a mixed pool of proximal and distal complexes). As we could not design a good strategy, we retained the fully spatially resolved microscopy-based approach.

5. Figure 8. The biochemical properties presented in Figure 8 do not indicate homologies between orthologs or provide the interesting information. Namely, the phylogenetic sequence conservation and tertiary structure. It would be more informative to show multiple sequence alignments with statistical score values. If AlphaFold2 is used then protein 3D fold superimpositions between orthologs could be shown.

Our primary intent with this figure was as a summary diagram, rather than providing more data, but more bioinformatic insight is certainly useful.

We have added additional information about the proximal/distal paralogous pairs, to address all of your queries:

We have added phylogenetic trees for the DC1, 2 and 4 (previously named 3) families from a gather across trypanosomatid reference genomes. This shows that the divergence of proximal/distal paralogs is ancestral to this lineage. This was added as a new supplemental figure, and referenced at the first time this new paralogous pair is discussed.

We have added percentage identity of the various paralogous pairs, measured from Clustal Omega sequence alignment. Note that we have not added the full multiple sequence alignment, as there was no reasonable way to do this within the available space. However, any future researcher can easily generate them from the clearly provided gene IDs.

We have added the predicted structure of the DC1, 2 and 4. Note that DC1 and 2 are essentially entirely alpha helix, which is why we previously only showed the pLDDT (in this case, high confidence) and secondary structure (in this case, entirely alpha helix) structures. Note that the DC4 structure is not very informative for potential function, the most similar experimentally determined structure to the globular N terminal domain is d-alanine-d-alanine ligase (PDB 8EVV), used in bacterial cell wall synthesis – ie. completely unrelated biochemistry.

We have added the structural alignment of the proximal and distal DC1, 2 and 4 paralogous pairs, aligned using the FATCAT algorithm to allow for twists between domains to improve alignment of multi-domain proteins. The FATCAT algorithm provides statistical scores for structure similarity, which we report in the updated figure, which shows highly significant structural similarities for the p/dDC1, 2 and 4 paralogous pairs.

Minor comments

1. Revise the “Protein sequence analysis” method section. It does not read well and there are many errors.

We have comprehensively rewritten this part, separating the candidate protein identification and ortholog mapping from the bioinformatic sequence analysis.

Many important details for reproducibility are missing in the “Genetic modification” method section. The sequences of sgRNA should be provided and a brief description of the design and PCR protocol should be added instead of just referring to other work.

We would like to emphasise that the references to previous work refer to the original methods paper that developed the protocol and provide extensive detail. Nonetheless, we have added a clear summary of the methodology, including sgRNA design strategy and PCR reaction mixture and thermocycle. We have added the sequences of all primers, including those used for sgDNA (encoding the sgRNA), in Table S2.

3. In the “Flagellar beating analysis” method section the computation protocol description is not clear “A range of beating characteristics were computed for each cell and all deletion cell lines were compared to the parental cell line. Full code for the analysis pipeline (including thresholds for exclusion) is available on request”. The script should be made available with a link (a requirement of this journal) so that readers can reproduce the analysis.

We have been updated the section to add detailed explanation of the beating characteristics that we used. The analysis scripts have been made available on Github, <https://github.com/Mar5bar/flagella-analysis-pipeline>, this has been added a reference to the methods section. These are the complete scripts necessary to go from videos through to the data we plotted.

4. Technical Details and Reproducibility. Ensure that all experimental procedures, especially those related to protein tagging and deletion generation, are described in sufficient detail for reproducibility. This includes specific oligo sequences used for tagging, details of PCR conditions, and any controls employed.

As noted above, we have extensively revised this section. All primer sequences, including those used for controls, are now provided in Table S2 and the PCR reaction mixtures and

thermocycles added to the text. As noted above, these match the single reference that we previously gave.

5. Typo in the introduction. "The outer dynein arms (ODAs) attached to the nine doublets every 24 nm and are canonically...". The outer dynein arms (ODAs) ARE attached...

Thank you, this has been corrected.

6. Typo in the introduction. "Overall, this suggests interplay of signalling and asymmetric protein distribution". Replace with "Overall, this suggests AN interplay of signalling ..."

This has been corrected.

7. Typo in the Results section. "However, the localisation of PDEB2::mNG, a representative distal enriched PDEs, was unaffected (Figure 3D)." Change to "a representative distal enriched PDE, was ..."

This has been replaced, as we have now carried out the experiments and show the data for both PDEB1 and 2.

Reviewer #2

Cecile Fort et al. present an insightful study on the proximal-distal asymmetry of the flagellum in *Leishmania mexicana*, a well-characterized flagellated parasite and a model system for examining flagellar assembly and function. Building on previous findings of proximal-distal asymmetry in the Outer Dynein Arm-docking complex (ODA-DC) within the flagellum of *Trypanosoma brucei* (Edwards et al. PNAS 2018), the authors utilized the *T. brucei* genome-wide protein tagging database TrypTag to identify 55 proteins with proximal-distal asymmetric localization.

Recognizing the importance of conserved components across species, these proteins were cross-referenced with orthologs in *L. mexicana*, retaining 15 proteins exhibiting comparable localization asymmetries in both species. These selected proteins underwent detailed analysis in *L. mexicana* to elucidate their roles in flagellum structure and function. By measuring the fluorescence signal distribution along the axoneme, distinct proximal-distal distribution patterns were revealed. Proteins with similar localization patterns were classified as proximal DC-like (pDC-like) or distal DC-like (dDC-like), and additional patterns such as "short proximal" and "distal enriched PDEs" were identified.

Through the generation of deletion mutants, the authors evaluated the role of these proteins in flagellar assembly and function, assessing flagellar length, ultrastructure via electron microscopy (EM) averaging densities, swimming behavior, and flagellar beat patterns. The study explored the dependency of proximal and distal-specific protein localization on DC asymmetry. It was found that proteins with pDC-like localization depended on the presence of pDC1 and pDC2, while short proximal proteins were unaffected by pDC1 deletion. Similarly, the deletion of dDC2 impacted the localization of dDC-like proteins but not distal enriched PDEs, indicating the presence of two distinct mechanisms for proximal and distal-specific localization in trypanosomatids. Interestingly, deletions of certain proximal-specific proteins affected the overall beat frequency without altering the structural integrity of the flagellum, suggesting a regulatory role in flagellar beat modulation.

Flagella and cilia are critical for the motility and sensory functions of many eukaryotic cells, including the pathogenic trypanosomatids *Trypanosoma brucei* and *Leishmania mexicana*. The structural and functional asymmetry of flagella is vital for generating effective beating patterns necessary for locomotion and host infection. Thus, understanding the molecular architecture of the flagellum is essential for elucidating the mechanisms underlying flagellar beating.

This comprehensive analysis of proximal-distal asymmetry in flagellar proteins between *T. brucei* and *L. mexicana* underscores the conservation and functional significance of these asymmetries. The study identifies distinct localization mechanisms and provides insights into the regulation of flagellar beating. These findings advance our understanding of the conservation and functional implications of these asymmetries across different species, enhancing our knowledge of flagellar biology.

Overall, this is an excellent article. The text is well-written, and the data are presented clearly and logically. The findings appear robust and support the conclusions. This work is highly significant for the broader community studying flagella and cilia structure and function, offering numerous hypotheses to be tested in other systems, particularly in patients with ciliopathies.

Major Comments:

The asymmetry in proximal-distal localization of proteins along flagellar axonemes has been previously described, notably in *Trypanosoma* by the authors, as well as in *Chlamydomonas* and humans. In *Chlamydomonas*, more extensive structural heterogeneities among microtubule doublets (MTDs) were observed in the proximal region of the axoneme, with one microtubule doublet exhibiting particularly strong proximal/distal asymmetry. The study presented here is already highly innovative, and incorporating recent advancements in super resolution microscopy could further enhance its novelty and impact, making it an even stronger candidate for high-impact journals such as *Nature Communications*. Expansion microscopy significantly enhances resolution and allows for high-resolution mapping of proteins relative to microtubule doublets. It could provide a more detailed understanding of protein localization within the axoneme, potentially uncovering new insights into the mechanisms of proximal-distal asymmetry. For example, recent studies by Laporte et al. *Cell*. 2024 have demonstrated the effectiveness of expansion microscopy in revealing intricate details of centriole assembly.

Expansion microscopy certainly has been powerful for centriole/basal body biology, and is a methodology that we considered. However, we rejected it as very unlikely to have sufficient resolution to be useful – as summarised nicely by a recent review (DOI: 10.1016/j.sbi.2023.102614) typical 4x expansion microscopy has a resolution of ~70 nm, increasing to ~30-40 nm with super resolution microscopy achievable at moderate throughput and not requiring specialised fluorophores (ie. SIM or Airyscan).

For axonemal structures, the most impressive expansion microscopy examples are from structures like transition fibres which project from the axoneme (like Cep89, DOI: 10.1016/j.cub.2022.12.046), and where transverse views of a shorter structure (ie. basal body/centriole) are possible. For the axoneme, there are not large external projections nor facile transverse sections.

Finally, we anticipated many proteins to be within the ODA/DC – borne out by identification of 10 as DC components or dependent on the DC and/or ODA for their localisation (p/dDC1/2/4, FLAM6, pDAP1, LC4-like, SPA1). The ODA/DC is a <20 nm structure, so far from resolvable by expansion microscopy.

While this study provides a systematic analysis of proteins with asymmetric localization in axonemes, it would benefit from a discussion on the role of this asymmetry across different stages of the parasitic life cycle. Specifically, *Leishmania* transitions into an intracellular amastigote form, which features a very short but motile flagellum, in contrast to *Trypanosoma brucei*, which lacks an intracellular stage. A brief discussion on this aspect would be particularly valuable for understanding the *L. mexicana* orthologs that do not exhibit asymmetry in their localization in the study (what about orthologs in *T. cruzi*, which also adopts an amastigote form with a short motile flagellum?). This could provide critical insights into the functional and evolutionary adaptations of flagellar proteins in relation to the distinct life cycle stages and cellular environments of *Leishmania* compared to *Trypanosoma brucei*.

This is an interesting point – by focusing on the proteins conserved from *T. brucei* to *L. mexicana* we have tried to focus on those most likely involved in highly conserved motility behaviours, but kinetoplastid parasite flagella do far more than just beat! We have added a short comment in the discussion on the complexity of flagellum adaptation through the parasite life cycles, and reflect on how lineage-specific adaptation asymmetry may be involved in non-motility functions.

Minor comments:

page 2: Correct the spelling in "including a new paralogous."

This sentence has been modified.

page 3:- Provide a table listing the 55 identified *T. brucei* proteins for easier reference.

All 55 *T. brucei* proteins were listed non-redundantly in Tables S1, S2 and S3 (organised in this way to match the content of Figure S1, S2 and S3). However, we can see that this was confusing, so we have consolidated Tables S1, S2 and S3 into a single table.

- Consider presenting the first three columns of Table S4 earlier in the manuscript and consistently use either the protein names or accession numbers throughout the text and figures. It would be much easier for the reader to track the various proteins.

We apologise for the lack of clarity, with detailed analysis of so many proteins we do recognise that it can be hard to track proteins through the figures. Through consolidation of the supplemental tables in to one table, this should now be clearer.

Our intent was to use gene IDs for all novel proteins and use gene names for all previously characterised proteins throughout the text and figures, up until the point in the discussion where we assign protein names. We have checked the text and figures to make sure that this has been done consistently. We have also updated Table S1 to add a column with the existing and new protein names, for easy cross-reference.

- Clarify "leaving 10 proteins for detailed analysis." The text suggests 15 proteins were retained, with 14 further characterized.

Apologies, this statement did gloss over detail. 15 proteins were listed in total. 5 of them (pDC1, pDC2, dDC1, dDC2 and LC4-like were studied by us in Edwards and al, 2018 specifically in the context of proximal-distal asymmetries, which left 10 proteins for detailed analysis. We have updated the text to simplify this statement.

- In the Table S2 legend, specify what "enriched proteins" means (compared to what?).

Commented [RW1]: Policy: Always show/say gene ID. For previously named proteins, always show existing name. For newly named proteins, only add protein name once named.

Here, we used enriched to enriched relative to the other end of the flagellum, although more precisely mean a gradually increasing gradient of signal - we excluded proteins from our analysis which localised to the entire axoneme but with enrichment at one end. As part of our efforts to improve clarity about protein localisation descriptions, we have stopped using this phrasing.

- For Figure S1, indicate whether the proteins were also N-terminal tagged to control for asymmetric localization.

We agree that it is very important to be clear when C and or N terminal tagging was done, and we have reformatted Figure S1, S2 and S3 to have consistent and clearer indication of the terminus of tagging.

Regarding your specific comment, tagging was almost always N and C terminal tagging in *T. brucei* and C terminal tagging in *L. mexicana*. This strategy was selected on the basis that N and C terminal tagging had very high agreement in *T. brucei*, with the most common problem being lack of fluorescent signal (rather than a different localisation) by N terminal tagging (4 examples). Building on this evidence that terminus of tagging did not affect localisation, we then tested C terminal tagging in *L. mexicana*. Between N and C in one species and (mostly) C terminal tagging in another species we argue that this is sufficient confidence.

This workflow has been clarified in the results, to justify our confidence in tagging only one terminus in *L. mexicana*. There is, of course, the possibility of aberrant localisation due to protein tagging (as highlighted by Reviewer 1 for Arl13B), and have clarified the text (mentioning it in both the results and the discussion) with this in mind.

- In Figure S4, confirm the deletion of LmxM.30.0090 because a higher band is observed and the PCR control (PF16) did not work.

You are absolutely correct, and we apologise for this oversight. As you note, this was an error interpreting the deletion validation PCR gel as we overlooked the size difference between the C9T7 gDNA/LmxM.30.0090 ORF primers control and the corresponding PCR for the cell line.

We have carefully checked all other PCR product sizes for the other cell lines – all do have the expected band sizes, so this was an isolated error.

We have replaced all the corresponding data with data from a confirmed deletion cell line: Figure 2B (flagellum length measurement) and 2F (electron microscopy imaging), and swimming/beating in 7A-C and the associated supplemental figure. This does show a phenotype that might be predicted, contributing to distal DC assembly, and we have been updated main text and the discussion correspondingly.

Because of this error, we sought additional experimental evidence about the function of pDC3 and dDC3. To do so, we generated the double deletion of pDC3 and dDC3, showing that this has a cumulative effect on the defect in ODA assembly and a corresponding defect in swimming – done in parallel to a double deletion of pDC1 and dDC2 as a comparison to the canonical DC components. This additional analysis has been added as Figure 2E and 7G, H, I.

On a related note, we received feedback from the posted preprint that pDC3 and dDC3 as protein names may cause confusion – this is because the *T. brucei/L. mexicana* p/dDC3s have no similarity to the previously characterised *C. reinhardtii* DC3. Therefore, we have changed the name of these proteins to p/dDC4.

page 4: - "As these included Δ pDC2, we believe these noisy observations are spurious, perhaps due to small variability in doublet position (Figure 2D, 2E)."

These observations could be influenced by factors not yet identified, and exploring these potential impacts may provide further insights."

This is possible, but we didn't see any patterns – which is why we draw the conclusion that this is some kind of noise. We will leave the full data available in the paper for future researchers to try to draw their own conclusion, and have rephrased the text to explain this possibility.

page 5: - PDEB2 localization was unaffected in the DC2 deletion mutant. The thorough analysis of PDEB2 localization is commendable. Additionally, assessing whether PDEB1 is affected could further elucidate the differential roles of PDEB1 and PDEB2, adding another layer of depth to this already insightful study.

As suggested, we have carried out this experiment with PDEB1 and have added this new PDEB1 data to Figure 2.

As might be expected from the high sequence similarity of these two proteins (percentage identity also now added to Figure S4D), both behave the same. Unfortunately, that leaves us unable to add any particular insight to any differential PDEB1 and 2 roles.

- In Figure 4A, consider reordering the images to present the Δ ODAA mutant first, as it aligns better with the text.

This is a nice suggestion to improve clarity, we have been updated the figure.

page 7: - Remove "that did" in "however, those that did had a significantly increased beat frequency."

We have been updated this sentence.

- For Figure 7B,E, note that Δ LC4-like, Δ PDEB2, and Δ FLAM6 display a higher proportion of cells with base-to-tip beating. Do the authors have a hypothesis for this observation?

We had noticed that the deletion mutants can have changes to prevalence of either base-to-tip, tip-to-base or both – although the changes are small compared to the large changes in propensity for tip-to-base beating. Given we were not able to quantitatively analyse the base-to-tip waveform, we think it is not appropriate to speculate on this weak result.

- For Figure 7C, clarify if the round dots represent the average. If so, Δ PDEB2 does not display a significant increase in beat frequency but more variability, suggesting less control.

The round dots do represent the average. We have corrected the figure legend to clearly state what the different features of the graph represent, and have updated the text to ensure that it is clear that PDEB2 deletion gives a high variability phenotype.

page 8

- Revise "We previously noted that ODA proximal-distal asymmetries occur across diverse eukaryotes." It would be insightful to discuss whether the characterized proteins in this study are conserved across other organisms and provide their IDs. This could enhance understanding of proximal-distal asymmetry in axonemes.

This links with comments by reviewer 1 about sensitive detection of orthologs of the proteins in this study, and we have carefully checked this using Jackhmmer. We have comprehensively revised Table S1 and include UniProt IDs of humans and *Chlamydomonas* ortholog IDs there. We have also carefully revised the related text in the discussion, including specifically listing the *Chlamydomonas* orthologs.

page 9:

- "The PFR-free and FAZ structures broadly line up with the short-proximal region, but dependence of PFR or FAZ positioning on axoneme asymmetries is unlikely as we saw no obvious change to the PFR or FAZ by electron microscopy in the SPA1 and SPA2 deletion mutants." The converse was not tested, so the conclusion should be toned down.

We agree that this conclusion can be toned down. As rightfully indicated, the converse has not been tested. Also, the morphology of *T. brucei* means that an association with the FAZ is not plausible. Given the wider suggestion of shortening the discussion, we have removed this discussion point.

page 10:

- Correct "We can consider a two key hypotheses" to "We can consider two key hypotheses."
- Correct the spelling of "increasesing"
- Remove "Domain identification, ortholog identification, structure prediction, etc. Cite tritrypdb, cite my alphafold."

These errors have been corrected

page 11:

- Correct "We used a positive control of with genomic DNA from the parental cell line" to "We used a positive control with genomic DNA from the parental cell line."
- Reformat "(Wheeler, 2017)."
- Correct the spelling of "1uL of 5 µm polystyrene beads."

These errors have been corrected

- Specify "Cells with only one observable flagellum" rather than "Cells with one flagellum (non-dividing)," as it is difficult to determine if a tiny new flagellum has not started to grow early in the cell cycle.

This has been rephrased as suggested

page 12:

- Specify the concentrations used for "Reynolds lead citrate71 (Lead nitrate (Thermofisher, L/1450), sodium citrate (Sigma, 71405) and sodium hydroxide (SLS, CHE3422)".

This information has been added to the text.

page 31

- Clarify the meaning of the asterisks in Table S4

The asterisk marks p and d DC1 and 2, because it is not entirely meaningful to say that these are DC-dependent as they are the core DC. These have been replaced with N/A.

Reviewer #3

In this strong and important paper, the authors complete a systematic identification of proteins with proximal-distal asymmetry in the axonemes of *Trypanosoma* and *Leishmania* species, indicating this phenomenon applies to more proteins than previously anticipated. Functional analysis and assessment of interdependencies among asymmetrically localized proteins identified at least two different mechanisms for achieving asymmetry, while examining a multitude of flagellum beat parameters demonstrated at least some of the proteins identified are required for normal flagellar beating.

The work is rigorous and the paper well-written. It represents an important contribution because, while longitudinal symmetries have been reported in several organisms, the extent of this phenomenon and underlying mechanisms remain poorly understood. Moreover, the requirement for such asymmetry in flagellar beating is poorly defined. This work therefore provides advances on multiple levels and provides a foundation for further dissecting the impact of and mechanisms for establishing longitudinal asymmetries in the axoneme. I have a few suggestions below for the authors that I think will improve the paper:

1. The DISCUSSION is a bit long and could be shortened, as this will improve the impact and reach of the paper, particularly for readers outside the direct area of flagellum biology.

We have carefully reviewed and shortened the discussion, focusing on points of more general interest outside of the immediate field and of broader interest in flagellar biology. Most points are still covered, but more concisely, with the exception noted above.

2. On p5 of 43, the sentence "...mutation of ...ODA4..." should be corrected as less general. Note that some mutations at the *oda4* (beta HC) alter ODAalpha, but others do not. e.g *J Cell Biol.* 1993 Aug 1; 122(3): 653–661.

Thank you, we agree that this is an important point to clarify. The key point of importance is that a truncated copy of the beta heavy chain, like in *oda4-s7*, may be sufficient for ODA assembly, while a complete loss of the protein, like in *oda4*, would not be expected to be sufficient for ODA assembly. We have carefully rephrased this section, including making it clear that in our *Leishmania* mutants we are doing complete deletions.

3. p7 of 43, last line of RESULTS.

-I advise removing the term "regulators", as this implies the protein may switch beat parameters in response to a signal. Results indicate loss of some proteins influences beat parameters, but a role as a "regulator" has not been assessed.

This is a very important clarification. Our intent was to reserve the term regulator for factors with clear potential as regulator, eg. cAMP-binding proteins, and be more nuanced when

presence or absence of a protein leads to a change in beat frequency. We have checked the text and updated use of this term.

4. Fig 7.

--please define "dominant" frequency.

The motion of the *L. mexicana* flagellum during tip-to-base symmetrical beats is almost entirely at a single frequency – this is the dominant frequency. We have greatly expanded the description of the methods for analysing the flagellum beat, which includes clear a definition of this term.

--The multi-factorial analysis of motility and flagellar beating is to be commended. Note, however, that it is hard to reconcile some of the swimming speed results with results of waveform analysis. For example, delta-pDC2 shows a dramatic increase in "uncoordinated" beating (now >60% of cells compared to ~5% in WT), yet no substantial change in swim speed. It is hard to imagine uncoordinated beating driving normal swimming speed. I suspect there is a fair amount of noise in the waveform analysis, so perhaps one can conclude general impact on flagellum beating, but care should be taken in drawing more specific conclusions. Please clarify/comment regarding this.

There are several factors which can lead to discrepancies here. While we would expect/hope a good correspondence between the low magnification swimming, medium magnification beat type and high magnification beat waveform analyses this is not guaranteed. There are three specific factors which may be either triggering different cellular behaviour or giving different observed behaviour – whether the cells are in a deep chamber or a shallow confined chamber, the light/heating which scales with microscope magnification, and observation biases (specifically, for beat analysis, only considering analysable cells doing a tip-to-base beat).

We have carefully rewritten the start of the results section which considers the motility data to make these potential sources of apparent discrepancy clear. We critically considered our conclusions in the light of this comment, and have rephrased some results.

5. I struggle a bit to distinguish "short proximal axoneme" from "pDC-like", as it seems both are defined as being in the proximal 20% of the axoneme. Please clarify.

These two localisations are ambiguous in *L. mexicana*, as the pDC region is short, however are readily distinguished in *T. brucei* due to the longer proximal flagellum region. In response to other comments, we have made numerous changes to these descriptions – which should clarify this point.

6. There is a recent paper examining mechanisms of proximal/distal asymmetry in trypanosoma that needs to be cited and discussed: Bonnefoy, S., Alves, A. A., Bertiaux, E. & Bastin, P. LRRC56 is an IFT cargo required for assembly of the distal dynein docking complex in Trypanosoma brucei. Mol. Biol. cell 35, ar106 (2024).

This is a very complementary study, providing direct evidence that the distal DCs are dependent on an IFT-carried protein for their normal assembly thus an IFT transport-based model for their asymmetry is plausible. We have added a brief description of this work to the discussion. Note that this was only not mentioned in the original submission as we believe it was not published at the time.

7. Fig 8.  in panel A, dDC is mislabeled as pDC

Thank you for spotting this error, we have corrected it.

Reviewer #1

Comments to the revised manuscript: Proteins with proximal-distal asymmetries in axoneme localisation control 1 flagellum beat frequency

The authors have made significant revisions based on the reviewers' feedback. To summarize.

They revised multiple figures, added a new supplemental figure, re-analyzed data, and incorporated double-deletion mutants. They expanded the methods section, introduced hierarchical clustering to strengthen conclusions on protein localization, and improved reporting of experimental details (e.g., oligonucleotide sequences and analysis pipeline code).

They employed enhanced ortholog detection using Jackhmmer but retained their original OrthoMCL/BLAST approach for sensitivity. In addition, They performed new quantitative analyses to differentiate protein localization in axonemal structures from transition zones.

Finally, they streamlined the discussion, clarified key points, and consolidated supplemental tables for improved clarity.

Thank you for acknowledging the efforts we have taken to improve the manuscript, we hope that you agree that these are significant improvements.

The authors have performed a thorough revision of their initial manuscript and added new data to support their conclusions and in some instances made corrections to the initial conclusions. I have the following major comments to the revised manuscript.

1. The authors reassessed the protein sequence homologies between *T. brucei* and *L. Mexicana* using Jackhmmer. In the rebuttal they state that Jackhmmer fails to detect several syntenic orthology pairs and decides to their existing method using a combined OrthoMCL/BLAST approach. To support their conclusion, they provide a list of syntenic pairs that Jackhmmer fails to detect. I have a problem with that analysis as I can see that several of the proteins provided in the list bear highly conserved and redundant protein domains (e.g., in the case of Tb927.10.5230, ARL13B) or bear large portions of coiled-coil regions (e.g., Tb927.11.16840 and other). In such cases, sensitive algorithms for phylogenetic analysis e.g., Jackhmmer or HHblits are not suitable as they produce much too many matches which makes phylogenetic analysis impossible. In that case, single BLAST searches based on overall sequence percentage homology are a better choice. Hence using Jackhmmer in this case does not prove any point. The HMM-based search tools are useful in all the other cases where only a few matches are found using BLAST. In those cases, usually, HHblits is the method of choice to expand the MSA and then convert that to an HMM that can be used to search against e.g., the proteome of an organism of choice. I suspect that if that sensitive approach was used (specifically in

those borderline cases) more matches would be found. This could be considered for future phylogenetic analysis.

Thank you for this clarification. We have carefully reevaluated our HMM-based analysis, which does indeed detect a few more potential *L. mexicana* orthologs of *T. brucei* proteins. However, and as would be expected, they are very low sequence similarity (typically <10%). Note the *T. brucei*-*L. mexicana* identity of ~40% for p/dDC1/2, despite their coiled-coil nature. Another useful benchmark is *T. brucei* vs *L. mexicana* Basalin (PMID: 30810527), which has 18.6% identity.

Our aim, as stated in the manuscript, was to identify and analyse proteins conserved among the trypanosomatid parasite family. If a particular protein is a “borderline case” for being detected as an ortholog, then it is not a good candidate for being – in a functional/biological sense – conserved fundamental biology of the trypanosomatid parasite flagellum.

As we have retained the BLAST/OrthoMCL definition of orthologs we have rephrased the conclusions on this point to emphasise that future more sensitive analysis may identify further more distant orthologs (Discussion first paragraph).

2. In the Figure 1E the authors present the new Hierarchical clustering of signal intensity profiles for flagellar asymmetrically distributed proteins. It was a bold claim in the initial version of the manuscript that four protein localization groups were identified considering that the groups were manually assigned. It is great to see that the authors employed hierarchical clustering of signal intensity profiles and found several new what I guess are “localization groups” (now 14 proximal and distal clusters identified) but now termed classes. But now that multiple groups/classes have been identified this begs the question. What is the physiological significance of all these clusters (groups)? Is it at all likely that so many compartments exist inside the flagellum? What if that twice the number of clusters (28) had been identified instead? Could the several clusters identified reflect the possibility that these intraflagellar proteins are spatially and temporarily dynamic rather than occupying specific localizations in flagellar? That is, do these many clusters reveal a weakness in the measuring approach? There should be some reflections about these concerns in the discussion. The authors note that many axoneme-localizing proteins localize along the entire flagellum as expected. However, these proteins are known not to be dynamic in terms of localization. It is another problem dealing with dynamic proteins that might switch places continuously.

Hierarchical clustering does not define a number of clusters. Defining number of clusters is mathematically not a well-defined problem, with numerous approaches which readily give conflicting answers. The groups that we indicated is essentially an arbitrary choice which was useful for describing the data.

This is not a claim for the number of clusters that exist, and we have clarified the text, avoiding the word 'cluster' and specifically referring to hierarchical similarity to emphasise this (Results paragraph 3).

The physiological relevance is explored through the data shown in Figures 3, 5, 6, where we demonstrate where there are biological dependencies for assembly, ie. functional consequences of the initial discovery and initial analysis presented in Figure 1. We have avoided referring back to similar localisation clusters/groups/hierarchical similarity unless there was functional support for them, ie. the pDC/dDC associated asymmetries.

I am afraid it is unclear what "spatially and temporarily dynamic" means here, or how it would make a protein have a characteristic sub-flagellar localisation. The most obvious problematic localisation would be intraflagellar transport particles, but these are readily recognisable. Ultimately, the data used for clustering represents the population average of the equilibrium state of any hypothetical dynamics – does this answer the question?

Please note that the statement that proteins localising along the entire length of the flagellum "are known not to be dynamic in terms of localization" is a statement of absence of evidence rather than one well supported by data. Indeed our previous analysis of the dDC/pDC asymmetry during flagellum growth shows that association of the ODAs with the axoneme must be dynamic on the time-scale of tens of minutes (PMID: 33093230). It may well be the case that other canonically stably associated structures have similar turnover. However, if we are interpreting this comment correctly, we believe this is tangential to our conclusions here.

3. I asked the authors to provide more biochemical evidence for their proposed intra cilia protein associations e.g., using affinity purification followed by proteomics analysis. In their response, the authors ask hypothetically "How would evidence for the interaction of LC4-like with an ODA heavy chain explain why this interaction only occurs in the distal flagellum? How would interaction of a novel protein with the proximal docking complex show that the novel protein is proximal specific, and preclude similar interaction with the distal docking complex?".

This biochemical evidence is important because most likely different cilia subcompartments reflect different subcomplexes as has been established for other cilia compartments e.g., the centrioles, basal body, transition zone, and the central pair. These scientific terminologies are justified because their existence is recapitulated by completely different methodologies i.e., imaging and biochemistry.

We appreciate your enthusiasm for including this additional data from an alternative methodology, however suggesting an alternative methodology is not the same as suggesting a viable experiment using that methodology.

We carefully considered pulldown approaches through the project and could not design experiments that would inform our primary conclusions. Please carefully re-read our previous comment and hypothetical example. We wrote that example as we cannot see how a positive or negative result for LC4-like pulldown of ODA heavy chains would change our key conclusions. A negative result may be because there is no interaction. However, we have electron microscopy evidence that LC4-like deletion causes specific loss of electron density in the distal ODAs and multiple cell biology experiments which show that LC4-like is dependent on expression of the ODAs and dDCs for its distal axonemal localisation. A negative pulldown result (easily explained by biological reasons like a weaker or dynamic interaction, an interaction labile on cell lysis, or an interaction which does not occur in a large pool of cytoplasmic ODAs pre-assembly, or technical reasons like steric masking of the epitope) would not supersede those positive electron microscopy and cell biological results for drawing our conclusions. More importantly, investing in this analysis would not inform anything about proximal-distal asymmetry, as we cannot spatially resolve from where material in the pulldown originated.

We acknowledge that this is one very basic hypothetical example. Perhaps you can see where a specific affinity purification experiment would provide key insight to support or refute our conclusions, however we cannot.

4. In their rebuttal, The authors state, "Because proteomic analysis of a proximity labeling, or affinity purification is not spatially resolved* it does not answer our primary question."

This is not true. Proximity labelling proteomics is in fact routinely used today in cell biology research exactly because it can detect spatially distinct protein niches/compartments in the cell. A publication in Nature Communications, I believe, requires evidence at the protein level to support any claim of a new "protein association" as suggested in the first version of the manuscript. This is more imperative now that the initial four "localization groups" have been ruled after more careful examination in the revised manuscript.

As for an affinity purification experiment, we cannot see how a proximity labelling experiment would inform the mechanisms underlying proximal-distal asymmetry generation. Again, suggesting a methodology is not the same as suggesting a viable experiment using that methodology which informs our conclusions.

Proximity labelling is not a spatially resolved method. It generates pseudo-1-dimensional data, the nearest effective/accessible distance between the prey proteins and the nearest tagged bait, averaged over the potentially multiple prey environments in the sample (eg. a cytoplasmic pool, proximal axoneme and distal axoneme). This is not sufficient to analyse the complex proximal-distal asymmetries which we observed:

Consider LC4-like proximity labelling, building on our existing evidence that it is an ODA-associated protein. Almost all (~80%) of ODA heavy chains will be very close to LC4-like molecules, in the dDC region, so will be very heavily labelled. All dDC will be close to LC4-like molecules, so very heavily labelled. A protein specific to the ODAs only at the extreme flagellar tip will be close to LC4-like molecules, so very heavily labelled. Even though the flagellum is a somewhat 1-dimensional structure you would not predict that LC4-like proximity labelling would give informative data, even without considering further complications like a pre-assembly cytoplasmic pool.

Again, this is one hypothetical example. Perhaps you can suggest a specific proximity labelling experiment which would help falsify our conclusions, but we again cannot.

For the related comment on four localisation groups, please see above for why our new analysis does not rule out four localisation groups.

Reviewer #2

Dear Editor,

Thank you for the opportunity to review the revised manuscript from C. Fort et al. entitled "Proteins with proximal-distal asymmetries in axoneme localisation control flagellum beat frequency".

I appreciate the authors' effort in addressing my comments. Overall, they have successfully tackled most of the issues I raised, and I believe the manuscript has improved as a result.

The authors have successfully clarified several points that were previously ambiguous and have provided additional data that strengthens their conclusions.

The manuscript has been restructured in parts to improve the flow and readability, which is beneficial.

Thank you for your constructive criticism and we are very pleased to have addressed your comments to your satisfaction.

Note : In Figure 2E and Figure 8: the panels are cut off and not fully visible. Table S3 is not mentioned in the text.

We acknowledge the issues with display of Figure 2 and 8, and apologise for this. Unfortunately, the submission system chooses to rotate and crops some figures in making the summary PDF, which is out of our control. The original files are correct and their display will be carefully checked at the proofing stage.

Apologies for not correctly referring to Table S3 in the text – we mistakenly referred to it as S2 rather than S3 and have corrected this mistake. (Materials and Methods > Electroporation and drug selection).

Aside from these minor issues, I am satisfied with the authors' revisions and recommend acceptance of the manuscript.

Reviewer #3:

The authors have provided a very thoughtful and thorough revision that addresses reviewer concerns. The paper is improved and a nice piece of work

Thank you, we very much appreciate your previous constructive feedback and are glad that you find our revisions suitable.